# Type VI secretion system killing by commensal *Neisseria* is influenced by expression of type four pili

**Rafael Custodio, Rhian M Ford[†], Cara J Ellison, Guangyu Liu, Gerda Mickute[‡], Christoph M Tang, Rachel M Exley***

Sir William Dunn School of Pathology, University of Oxford, Oxford, United Kingdom

**Abstract** Type VI Secretion Systems (T6SSs) are widespread in bacteria and can dictate the development and organisation of polymicrobial ecosystems by mediating contact dependent killing. In *Neisseria* species, including *Neisseria cinerea* a commensal of the human respiratory tract, interbacterial contacts are mediated by Type four pili (Tfp) which promote formation of aggregates and govern the spatial dynamics of growing *Neisseria* microcolonies. Here, we show that *N. cinerea* expresses a plasmid-encoded T6SS that is active and can limit growth of related pathogens. We explored the impact of Tfp on *N. cinerea* T6SS-dependent killing within a colony and show that pilus expression by a prey strain enhances susceptibility to T6SS compared to a non-piliated prey, by preventing segregation from a T6SS-wielding attacker. Our findings have important implications for understanding how spatial constraints during contact-dependent antagonism can shape the evolution of microbial communities.

**\*For correspondence:**
rachel.exley02@path.ox.ac.uk

**Present address:** [†]School of Biosciences, Sutton Bonington Campus, University of Nottingham, Nottingham, United Kingdom; [‡]MRC Weatherall Institute of Molecular Medicine, University of Oxford, Oxford, United Kingdom

## Introduction

The human microbiota is critical for the development of a healthy gastrointestinal immune system (*Round and Mazmanian, 2009*; *Sommer and Bäckhed, 2013*) and can also protect the host from invasion by pathogenic bacteria (*Kamada et al., 2013*). The microbes that carry out these important functions live as part of complex communities shaped by their fitness and ability to adapt to their environment, and which can be remodeled through mutualistic and antagonistic interactions (*García-Bayona and Comstock, 2018*; *Little et al., 2008*; *Nadell et al., 2016*). Competition for niche and host-derived resources has therefore driven the evolution in bacteria of an array of mechanisms to suppress growth of or kill neighbouring microbes. One mechanism, the Type VI Secretion System (T6SS), provides an effective strategy to eliminate competitors in a contact-dependent manner and is widespread in Gram negative bacteria from many different environments (*Coulthurst, 2019*). The T6SS is a contractile, bacteriophage-like nanomachine that delivers toxins into target organisms (*Cianfanelli et al., 2016*; *Ho et al., 2014*). T6SS-associated effectors possess a broad range of activities, including nucleases (*Koskiniemi et al., 2013*; *Ma et al., 2014*; *Pissaridou et al., 2018*), phospholipases (*Flaugnatti et al., 2016*; *Russell et al., 2013*), peptidoglycan hydrolases (*Whitney et al., 2013*), and pore-forming proteins (*Mariano et al., 2019*); each effector is associated with a cognate immunity protein to prevent self-intoxication and to protect against kin cells (*Alcoforado Diniz et al., 2015*; *Unterweger et al., 2014*). In pathogens such as *Pseudomonas*, *Vibrio*, *Salmonella*, and *Shigella*, the impact of the T6SS in pathogenesis and bacterial competition has been established in vitro and in some cases in vivo (*Anderson et al., 2017*; *Sana et al., 2016*). Commensal bacteria also harbour T6SSs, although how these systems combat pathogens has only been elucidated for Bacteroidetes in the intestinal tract (*Russell et al., 2014*); further studies are needed to gain a greater

appreciation of how T6SSs in commensals influence microbial communities and pathogens in other niches.

The human nasopharynx hosts a polymicrobial community (*Kumpitsch et al., 2019*; *Cleary and Clarke, 2017*; *Ramos-Sevillano et al., 2019*), which can include the obligate human pathogen *Neisseria meningitidis*, as well as related but generally non-pathogenic, commensal *Neisseria* species (*Diallo et al., 2016*; *Dorey et al., 2019*; *Gold et al., 1978*; *Knapp and Hook, 1988*; *Sheikhi et al., 2015*). In vivo studies have demonstrated an inverse relationship between carriage of commensal *Neisseria lactamica* and *N. meningitidis* (*Deasy et al., 2015*), whereas in vitro studies have revealed that some commensal *Neisseria* demonstrate potentially antagonistic effects against their pathogenic relatives (*Custodio et al., 2020*; *Kim et al., 2019*). Commensal and pathogenic *Neisseria* species have also been shown to interact closely in mixed populations (*Custodio et al., 2020*; *Higashi et al., 2011*). Social interactions among *Neisseria* are mediated by surface structures known as Type IV pili (Tfp). These filamentous organelles enable pathogenic *Neisseria* to adhere to host cells (*Nassif et al., 1993*; *Virji et al., 1991*) and are crucial for microbe-microbe interactions and the formation of bacterial aggregates and microcolonies (*Helaine et al., 2007*; *Higashi et al., 2007*). In addition, Tfp interactions can dictate bacterial positioning within a community; non-piliated strains have been shown to be excluded to the expanding edge of colonies growing on solid media (*Oldewurtel et al., 2015*; *Zöllner et al., 2017*) while heterogeneity in pili, for example through post translational modifications, can alter how cells integrate into microcolonies (*Zöllner et al., 2017*).

*Neisseria cinerea* is one of the commensal *Neisseria* species that has been previously isolated from the upper respiratory tracts of adults and children (*Knapp and Hook, 1988*; *Sheikhi et al., 2015*). This species expresses Tfp that promote microcolony formation, can closely interact with *N. meningitidis* in a Tfp-dependent manner and impairs meningococcal association with human epithelial cells (*Custodio et al., 2020*). Here, whole genome sequence analysis revealed that the *N. cinerea* isolate used in our studies encodes a T6SS. Similarly, T6SS genes have been recently identified in other *Neisseria* spp. isolated from human throat swab cultures (*Calder et al., 2020*). Here, we provide the first description of a functional T6SS in *Neisseria* spp. We show that the *N. cinerea* T6SS is encoded on a plasmid and antagonises pathogenic relatives, *N. meningitidis* and *Neisseria gonorrhoeae*. Moreover, we examined whether Tfp influence the competitiveness of microbes in response to T6SS-mediated antagonism and demonstrate that T6SS–mediated competition is facilitated by Tfp in bacterial communities.

## Results

### N. cinerea 346T encodes a functional T6SS on a plasmid

We identified a single locus in *N. cinerea* isolate CCUG346T (346T) (https://www.ccug.se/strain?id=346) that encodes homologues of all 13 components that are necessary for a functional T6SS (*Cascales and Cambillau, 2012*), including genes predicted to encode canonical T6SS components Hcp and VgrG (*Figure 1A* and *Supplementary file 1*). We used T6SS-effector prediction software tools (*Li et al., 2015*) to search for putative effectors. In total we identified six putative effector and immunity genes, termed *nte* and *nti* for Neisseria T6SS *e*ffector/*i*mmunity, respectively.

Of note, all predicted T6SS-related *orf*s and Nte/Ntis in *N. cinerea* 346T were found to be encoded on a 108,141 bp plasmid, revealed by PacBio sequencing, and confirmed by PCR and sequencing. Nte/Nti 1–5 are encoded adjacent to the structural gene cluster, with Nte6/Nti6 encoded elsewhere in the plasmid (*Figure 1B* and *Figure 1—figure supplement 1*). No other PAAR, Rhs, or VgrG homologues were found outside the plasmid. Thus, our analysis reveals that the human commensal *N. cinerea* 346T harbours a plasmid-borne T6SS together with six putative effector-immunity pairs.

Contraction of the T6SS leads to Hcp secretion, a hallmark of a functional T6SS (*Cascales and Cambillau, 2012*). Therefore, to establish whether the *N. cinerea* T6SS is functional, we assessed Hcp levels in whole cell lysates and supernatants from wild-type *N. cinerea* and a ΔtssB mutant, based on previous work demonstrating that TssB is a component of the T6SS-tail-sheath required for contraction (*Brackmann et al., 2018*). A ΔT6SS mutant which lacks 10 core genes including *hcp* was analysed as a negative control. As expected, Hcp was detected in both fractions from the wild-type strain but not in the negative control strain (ΔT6SS mutant) (*Figure 1C*). Importantly, Hcp was

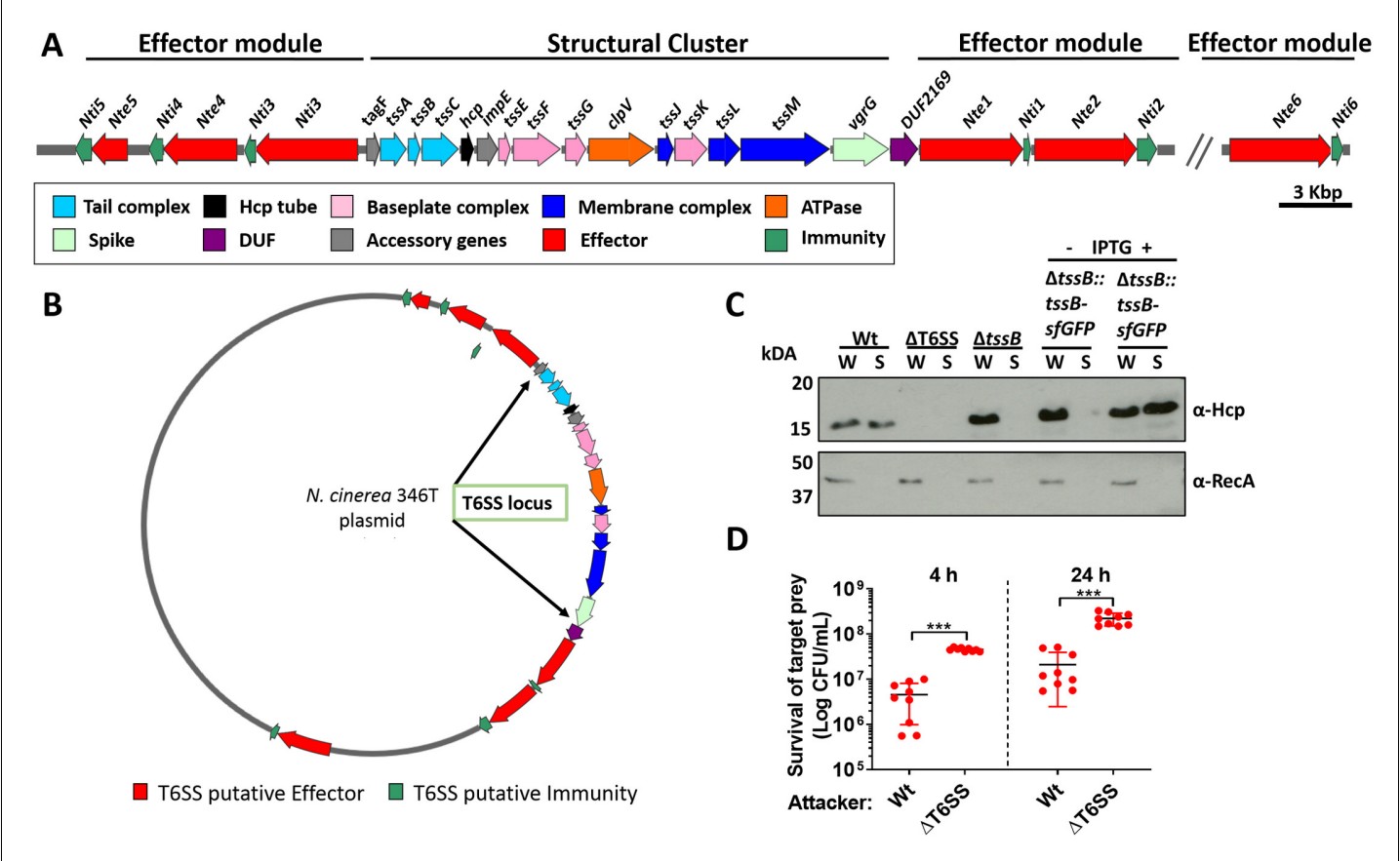

**Figure 1.** *N. cinerea* expresses a functional T6SS. (**A**) Schematic representation of T6SS genes in *N. cinerea* 346T. Canonical *tss* nomenclature was used for genes in the T6SS cluster. (**B**) Map of the T6SS-associated genes encoded by the *N. cinerea* 346T plasmid. See also *Figure 1—figure supplement 1*. (**C**) Expression and secretion of Hcp by wild-type *N. cinerea* 346T (Wt) and the *tssB* mutant (Δ*tssB*). Hcp protein was detected in the whole cell lysates (W) and supernatants (S) by western blot analysis. For strain Δ*tssB::tssB*sfGFP, bacteria were grown in the presence (+) or absence (-) of 1 mM IPTG; molecular weight marker shown in kDa. RecA is only detected in whole cell lysates. (**D**) Survival of the prey, *N. cinerea* 27178A, after 4 and 24 h co-incubation with wild-type *N. cinerea* 346T or the T6SS mutant (ΔT6SS) at approximately 10:1 ratio, attacker:prey. The mean ± SD of three independent experiments is shown: ***p < 0.0001 using unpaired two-tailed Student's t-test.

The online version of this article includes the following source data and figure supplement(s) for figure 1:

**Source data 1.** Western Blot of *N. cinerea* Hcp secretion and expression.

**Source data 2.** Survival of *N. cinerea* 27178A (prey) after 4 and 24 h competition with wild-type *N. cinerea* 346T or the T6SS mutant.

**Figure supplement 1.** The *N. cinerea* 346T T6SS is encoded on a plasmid.

present in cell lysates from the Δ*tssB* mutant, but not detected in cell supernatants, while Hcp secretion was restored by complementation of the Δ*tssB* mutant by chromosomal expression of TssB with a C-terminal sfGFP fusion (Δ*tssB::tssB*-sfGFP) (*Figure 1C*).

Next, we performed competition assays between *N. cinerea* 346T or the ΔT6SS mutant against *N. cinerea* 27178A which lacks a T6SS and Nte/Nti pairs identified in *N. cinerea* 346T. The survival of *N. cinerea* 27178A was reduced by around an order of magnitude following incubation with *N. cinerea* 346T compared with the ΔT6SS mutant (*Figure 1D*), confirming that the *N. cinerea* 346T T6SS is active during inter-bacterial competition.

## Dynamic behaviour of the *Neisseria* T6SS in the presence of prey cells

We further analysed the activity of the T6SS by visualising assembly and contraction in *N. cinerea* 346TΔ*tssB::tssB*-sfGFP; this strain exhibits comparable T6SS killing as wild-type *N. cinerea* 346T (*Figure 2—figure supplement 1*). Time-lapse microscopy revealed dynamic T6SS foci inside bacteria, with structures extending/contracting over seconds (*Figure 2A* and *Figure 2—video 1*) consistent

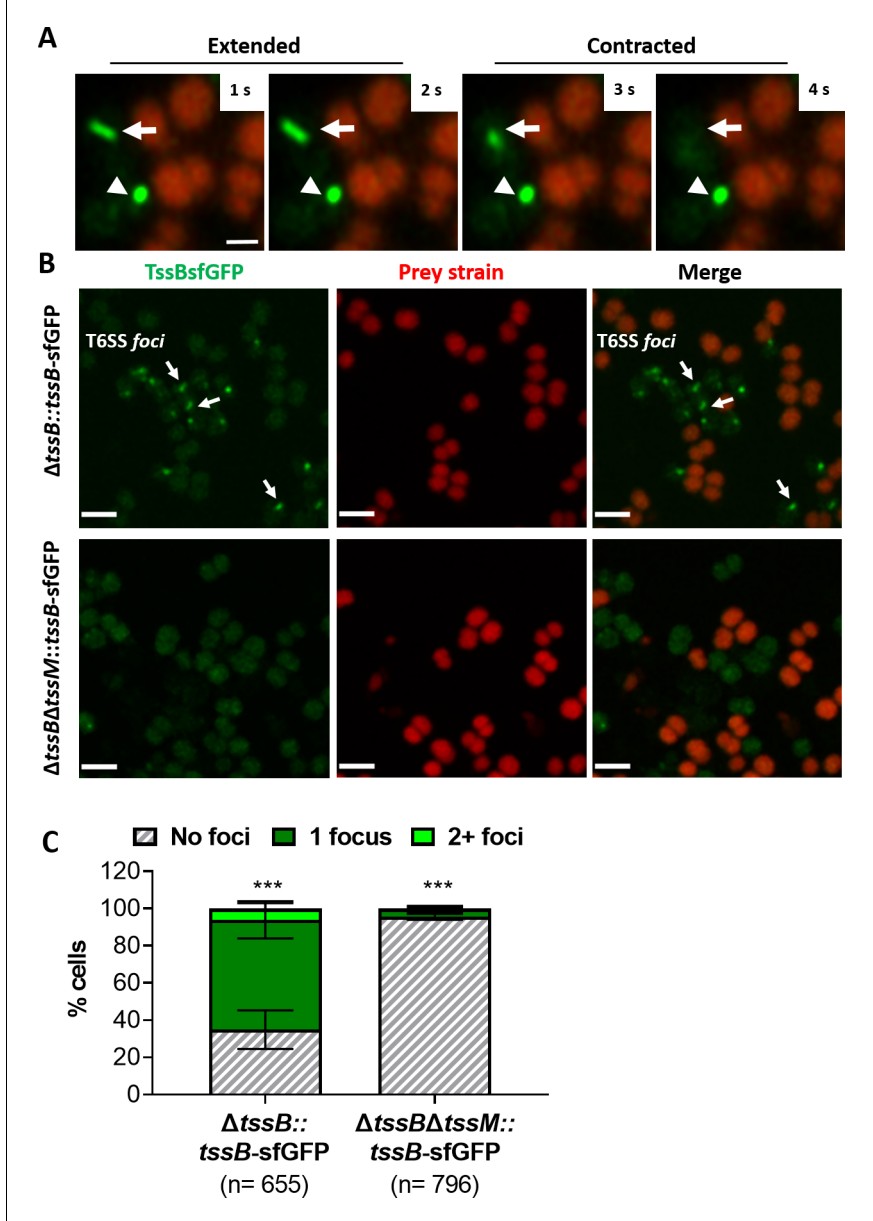

**Figure 2.** Visualisation of T6SS activity in *N. cinerea*. (**A**) Assembly and contraction of the T6SS in *N. cinerea*; white arrows indicate contracting T6SSs. Time-lapse images of *N. cinerea* 346T*ΔtssB::tssB*sfGFP (green) and prey *N. cinerea* 27178A_ *sfCherry* (red); the arrowhead shows a non-dynamic focus, scale bar, 1 µm. See also *Figure 2—video 1*. (**B**) Representative images of *N. cinerea* strains with the TssB::sfGFP fusion with (upper panels) or without (lower panels) TssM. Loss of fluorescent foci upon deletion of *tssM* indicates that foci correspond to active T6SSs. The scale bar represents 2 µm. (**C**) Quantification of TssB-sfGFP foci in different strains. T6SS foci were quantified using 'analyse particle' (Fiji) followed by manual inspection. For each strain, at least two images from gel pads were obtained on two independent occasions. Percentage of cells with 0, 1, or 2+ foci are shown and n = number of cells analysed. Data shown are mean ± SD of two independent experiments: ***p<0.0001 using two-way ANOVA test for multiple comparison. See also *Figure 2—video 2*.

The online version of this article includes the following video, source data, and figure supplement(s) for figure 2:

**Source data 1.** Quantification of TssB-sfGFP foci by live-microscopy.

**Figure supplement 1.** *N. cinerea* T6SS with a TssB C-terminal sfGFP fusion is functional and activity is lost upon deletion of *tssM*.

**Figure 2—video 1.** Visualisation of *N. cinerea* T6SS contraction.

https://elifesciences.org/articles/63755#fig2video1

*Figure 2 continued on next page*

*Figure 2 continued*

**Figure 2—video 2.** Visualisation of *N. cinerea* T6SS foci.

https://elifesciences.org/articles/63755#fig2video2

with T6SS activity (*Gerc et al., 2015*; *Ringel et al., 2017*). To further confirm T6SS activity, we deleted the gene encoding the TssM homologue in strain 346T*ΔtssB::tssB*-sfGFP, abolishing T6SS activity (*Figure 2B* and *Figure 2—figure supplement 1*) and confirmed that in the absence of TssM, fluorescent structures were rarely seen (< 5% of cells in the Δ*tssM* background, compared with > 60% in the strain expressing TssM; *Figure 2C* and *Figure 2—video 2*).

Finally, we examined whether T6SS assembly induces lysis of prey cells. We imaged *N. cinerea* 346T*ΔtssB::tssB*-sfGFP with *N. cinerea* 27178 expressing sfCherry on gel pads with SYTOX Blue as an indicator of target cell permeability (*Ringel et al., 2017*). Interestingly, we detected increased SYTOX staining of prey cells immediately adjacent to predator bacteria displaying T6SS activity (*Figure 3* and *Figure 3—video 1*), indicating that the *N. cinerea* T6SS induces cell damage and lysis of its prey.

## *N. cinerea* T6SS effectors are functional toxin/immunity pairs

To characterise the six putative T6SS effectors identified, we first used sequence analysis to determine their predicted domain structure. As shown in *Figure 4*, all Ntes contain a conserved Rhs domain, frequently associated with polymorphic toxins (*Busby et al., 2013*), and a C-terminal region with predicted activities previously described in T6SS effectors (*Alcoforado Diniz et al., 2015*). Nte1 contains an N-terminal PAAR motif, which can associate with the VgrG tip of the T6SS

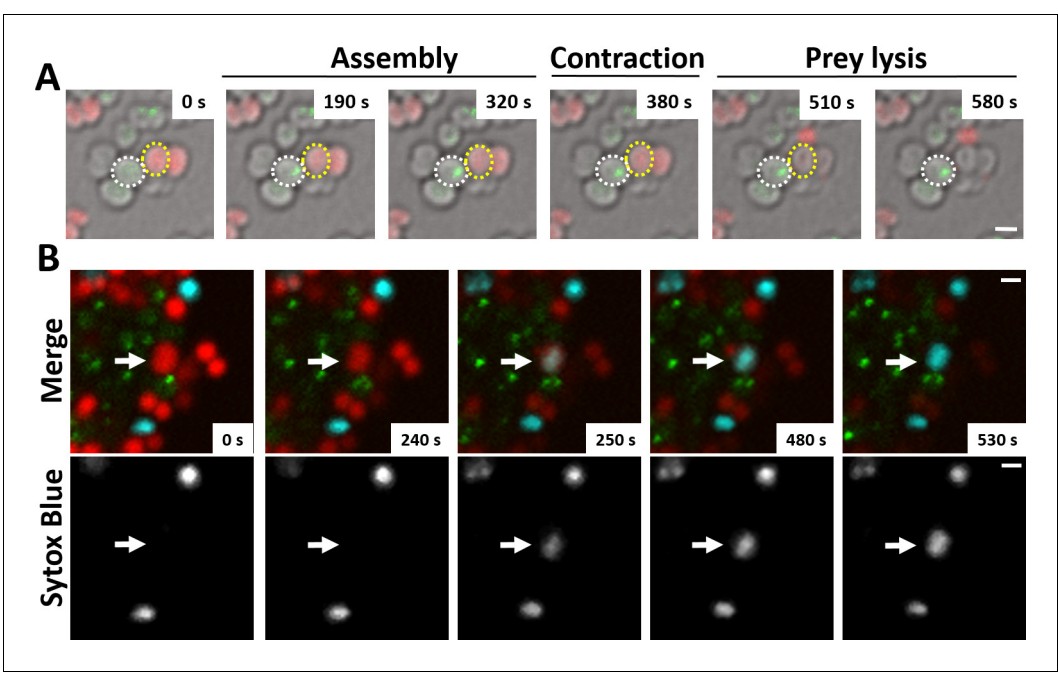

**Figure 3.** *N. cinerea* T6SS induces lysis in prey bacteria. (**A**) Assembly of T6SSs and prey lysis. Time-lapse series of merged images with phase contrast, *N. cinerea* 346T Δ*tssB+tssB*sfGFP (green), and *N. cinerea* 27178A sfCherry (red); scale bar, 1 µm. (**B**) Top row shows merged images of GFP (green, indicating T6SS assembly/contraction), mCherry (red, prey strain), and SYTOX Blue (cyan, showing membrane permeabilisation) channels. The bottom row arrows highlight a prey cell losing membrane integrity (increase in SYTOX Blue staining inside cells) arrows. Representative image from two biological repeats. Scale bars represent 1 µm. See also *Figure 3—video 1*.

The online version of this article includes the following video for figure 3:

**Figure 3—video 1.** *N. cinerea* T6SS elicits prey lysis.

https://elifesciences.org/articles/63755#fig3video1

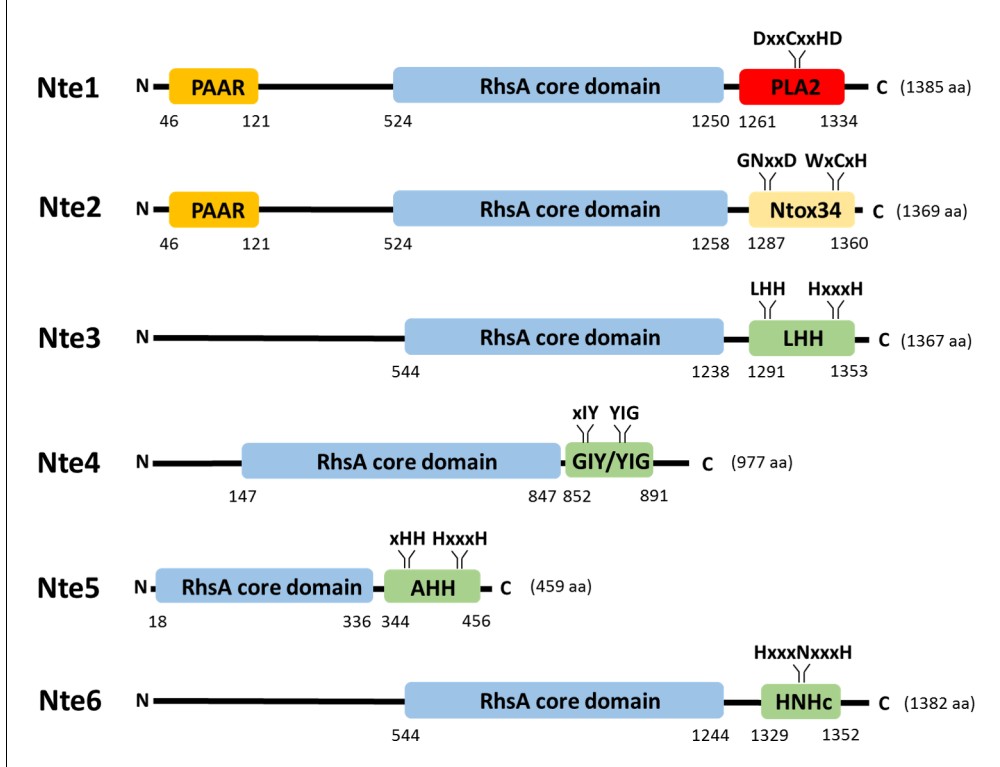

**Figure 4.** Predicted domain organisation of *N. cinerea* 346T T6SS effectors. Schematic representation of bioinformatically identified effectors in *N. cinerea* 346T. The domain organisation of the putative effectors is shown, with PAAR motifs indicated in orange, Rhs domains in blue, endonuclease motifs (Tox-LHH pfam14411; Tox-GIY/YIG cd00719; Tox-AHH pfam14412; and Tox-HNHc cd00085) in green, RNase (Ntox34, pfam15606) motif in yellow and the phospholipase (PLA2_like, cd00618) domain in red. The conserved domains annotation was retrieved from the NCBI database.

(*Shneider et al., 2013*) and C-terminal phospholipase domain (cd00618). Nte2 also contains an N-terminal PAAR domain and has a predicted RNase domain (pfam15606) in its C-terminal region. Using BLASTp analysis and the PAAR-like domain sequence from Nte1 as the query sequence, we did not identify any other PAAR encoding genes in the WGS of 346T. Nte3 is a putative endonuclease of the HNH/Endo VII family with conserved LHH (pfam14411). Nte4 contains a GIY-YIG nuclease domain (cd00719) and Nte5 is predicted to be an HNH/endo VII nuclease with conserved AHH (pfam14412), with Nte6 predicted to contain an HNHc endonuclease active site (cd00085).

To further characterise the possible effector/immunity pairs, we expressed each Nte alone or with its corresponding Nti using an inducible expression plasmid in *E. coli* (*Figure 5*). We were only able to clone wild-type Nte6 in presence of its immunity protein, so Nte6$^{R1300S}$ was used to analyse toxicity of this protein. In addition, as Nte1 encodes a predicted phospholipase that should be active against cell membranes (*Flaugnatti et al., 2016*), we targeted the putative phospholipase domain of Nte1 to the periplasm by fusing it to the PelB signal sequence (*Singh et al., 2013*); cytoplasmic expression of the Nte1 phospholipase domain does not inhibit bacterial growth (*Figure 5—figure supplement 1*). All Ntes are toxic, with their expression leading to decreased viability and reduced optical density (OD) of *E. coli* cultures compared to empty vector controls; toxicity was counteracted by co-expression of the corresponding Nti.

## Commensal *Neisseria* T6SS kills human pathogens

We next investigated whether *N. cinerea* can deploy the T6SS to antagonise the related pathogenic species, *N. meningitidis* and *N. gonorrhoeae*. We performed competition assays with three *N. meningitidis* strains (belonging to different lineages and expressing different polysaccharide capsules i. e., serogroup B or C), and a strain of *N. gonorrhoeae*. *N. cinerea* 346T caused between a 50- to

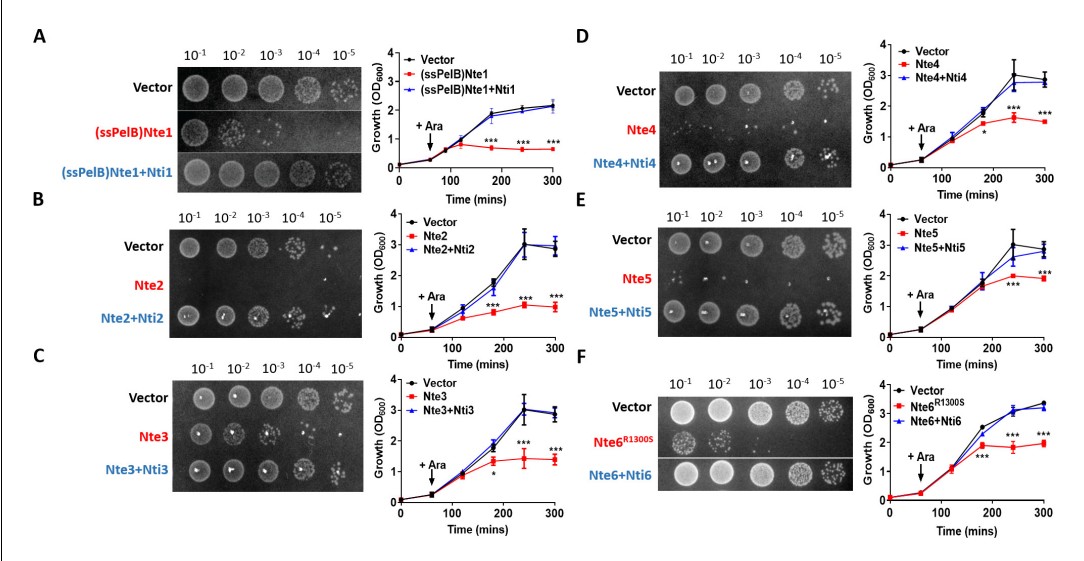

**Figure 5.** Putative *N. cinerea* T6SS effectors are toxic in *E. coli*. (**A**) Arabinose (Ara)-induced expression of T6SS effector Nte1 in periplasm of *E. coli* leads to reduction in CFU and OD at 600 nm ($OD_{600}$). Co-expression of putative immunity Nti1 restores growth to levels of strain with empty vector (pBAD33). See also *Figure 5—figure supplement 1*. (**B-E**) Cytoplasmic expression of putative effectors Nte2-5 without cognate immunity reduces growth and survival of *E. coli*. (**F**) Expression of Nte6[R1300S] reduces viability and growth when expressed in *E. coli*. Expression of Nti6 with Nte6 does not impact growth. In (**A-F**) number of CFU at 120 min post-induction are shown. Data shown are the mean ± SD of three independent experiments: NS, not significant, ***p<0.0001, *p<0.05 using two-way ANOVA test for multiple comparison. Images of colonies for Nte1 and Nte6 are composite as strains were spotted to different areas of the same plates.

The online version of this article includes the following source data and figure supplement(s) for figure 5:

**Source data 1.** Growth of *E. coli* strains expressing putative *N. cinerea* 346T effector/immunity .

**Figure supplement 1.** *N cinerea* putative T6SS effector Nte1 requires a PelB signal sequence for toxicity in *E. coli*.

100-fold decrease in survival of the meningococcus compared with the ΔT6SS strain, irrespective of lineage or serogroup (*Figure 6A*) and an approximately fivefold reduction in survival of the gonococcus (*Figure 6B*). We also investigated whether the meningococcal capsule protects against T6SS assault. Using a capsule-null strain (Δ*siaD*) in competition assays with wild-type *N. cinerea* 346T or the T6SS mutant, we found reduced survival of the unencapsulated mutant compared to the wild-type (*Figure 6C*). Therefore, the meningococcal capsule protects bacteria against T6SS attack.

## Spatial segregation driven by type IV pili dictates prey survival against T6SS assault

Despite the potency of the T6SS in *Neisseria* warfare, this nanomachine operates when bacteria are in close proximity, so we hypothesised that Tfp, which are critical for the formation of *Neisseria* microcolonies and organisation of bacterial communities (*Higashi et al., 2007*; *Mairey et al., 2006*; *Oldewurtel et al., 2015*; *Zöllner et al., 2017*), could influence T6SS-mediated antagonism. To test this, we constructed fluorophore expressing 'prey' strains (i.e. sfCherry-expressing 346TΔ*nte/i3-5*; *Figure 7—figure supplement 1*) with and without Tfp. Prey strains were mixed with piliated attacker strain *N. cinerea* 346T expressing sfGFP at a 1:1 ratio on solid media, and the spatiotemporal dynamics of bacterial growth examined by time-lapse stereo microscopy over 24 hr, while the relative proportion of each strain was analysed by flow cytometry at 24 hr (*Figure 7—figure supplement 2*). As expected based on previous observations of Tfp-mediated cell sorting in *Neisseria* (*Oldewurtel et al., 2015*; *Zöllner et al., 2017*), the non-piliated prey strain (346TΔ*nte/i3-5*Δ*pilE1/ 2_sfCherry*; red) segregates to the periphery of the colony, in this location the prey strain escapes T6SS-mediated assault and dominates the expanding colony (*Figure 7A* and *Figure 7—video 1*). In contrast, when the prey is piliated, pilus-mediated cell interactions prevent displacement of cells to the expanding front (*Oldewurtel et al., 2015*; *Pönisch et al., 2018*; *Zöllner et al., 2017*), so the

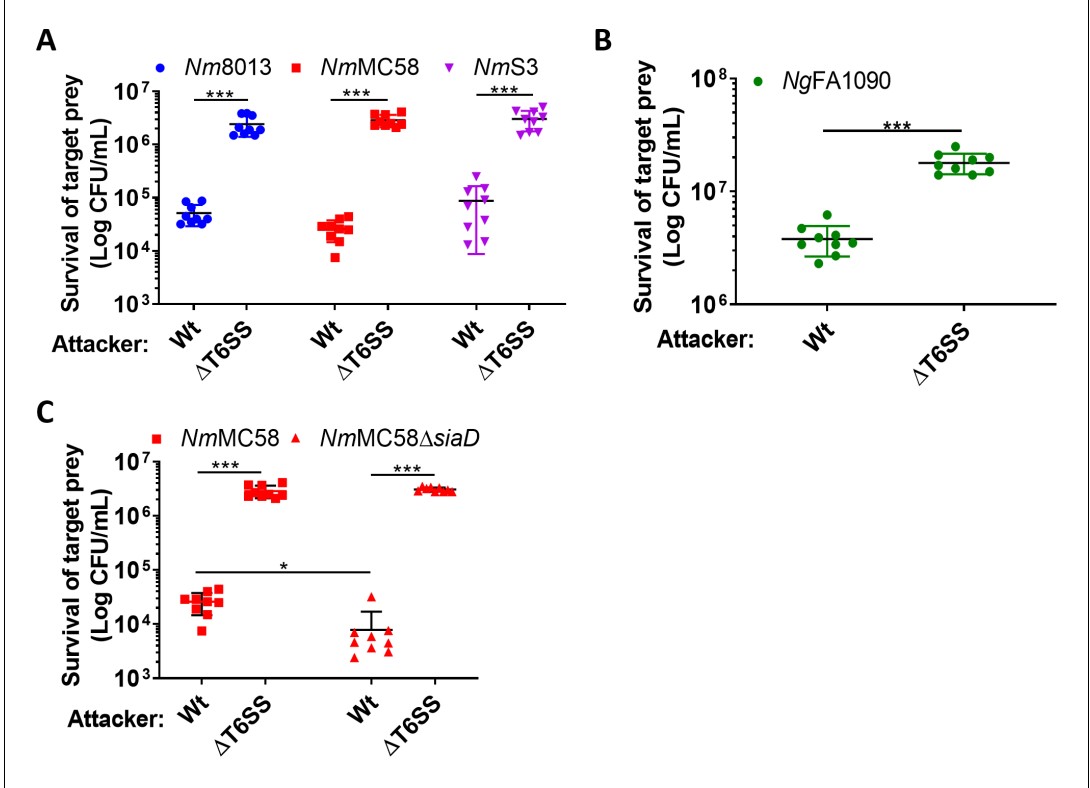

**Figure 6.** *N. cinerea* T6SS is active against pathogenic *N. meningitidis* and *N. gonorrhoeae*. (**A**) Recovery of wild-type *N. meningitidis* (*Nm*8013, *Nm*MC58, *Nm*S3) after 4 hr co-incubation with *N. cinerea* 346T wild-type (Wt) or the T6SS mutant (ΔT6SS) at approx. 100:1 attacker:prey ratio. (**B**) Recovery of wild-type *N. gonorrhoeae* (FA1090) after 4 hr co-incubation with *N. cinerea* 346T wild-type (Wt) or the T6SS mutant (ΔT6SS) at approximately 10:1 attacker:prey ratio. (**C**) Unencapsulated *N. meningitidis* (*Nm*MC58Δ*siaD*) is more susceptible to T6SS-mediated killing than wild-type *N. meningitidis*. Recovery of *Nm*MC58 or the capsule-null mutant (*Nm*MC58Δ*siaD*) after 4 hr co-culture with *N. cinerea* 346T (Wt) or a T6SS-deficient mutant (ΔT6SS) at ratio of approximately 100:1, attacker:prey. Data shown are the mean ± SD of three independent experiments: NS, not significant, ***p < 0.0001, **p < 0.001 using unpaired two-tailed Student's t-test for pairwise comparison (**B** and **C**) or one-way ANOVA test for multiple comparison (**A**).

The online version of this article includes the following source data for figure 6:

**Source data 1.** Survival of *N. meningitidis* strains after 4 hr co-incubation with *N. cinerea* 346T wild-type or the T6SS mutant.
**Source data 2.** Survival of *N. gonorrhoeae* FA1090 strain after 4 hr co-incubation with *N. cinerea* 346T wild-type or the T6SS mutant.
**Source data 3.** Survival of *N. meningitidis* MC58 or the capsule-null mutant strain after 4 hr co-incubation with *N. cinerea* 346T wild-type or the T6SS mutant.

susceptible strain (Tfp-expressing 346TΔ*nte*/*i3-5_sfCherry* Tfp+, red) is outcompeted by the T6SS+ strain (Tfp-expressing 346T_*sfGfp* Tfp+, green) (**Figure 7B** and **Figure 7—video 2**). When both strains are piliated and immune to T6SS attack, there is no dominance of either strain (**Figure 7C** and **Figure 7—video 3**). Assessment of the relative recovery of piliated and non-piliated prey in competition assays also supported the observation that the piliation status of the prey impacts survival against T6SS (**Figure 7D** and **Figure 7—figure supplement 3**). These results highlight that Tfp influence the outcome of T6SS-mediated antagonism through structuring and partitioning bacteria in mixed microcolonies.

We also considered that Tfp might contribute to increased prey survival through mechanisms other than the spatial organisation of strains within bacterial colonies. For example, Tfp-Tfp interactions are known to contribute to kin recognition (**Adams et al., 2019**) and promote aggregation (**Hélaine et al., 2005**), which could impact T6SS activity by anchoring neighbouring cells in closer proximity. Alternatively, Tfp may have a role in provoking T6SS activity, similar to the T6SS response to exogenous T6SS (**Basler et al., 2013**) or cell envelope perturbations in *Pseudomonas* (**Ho et al.,**

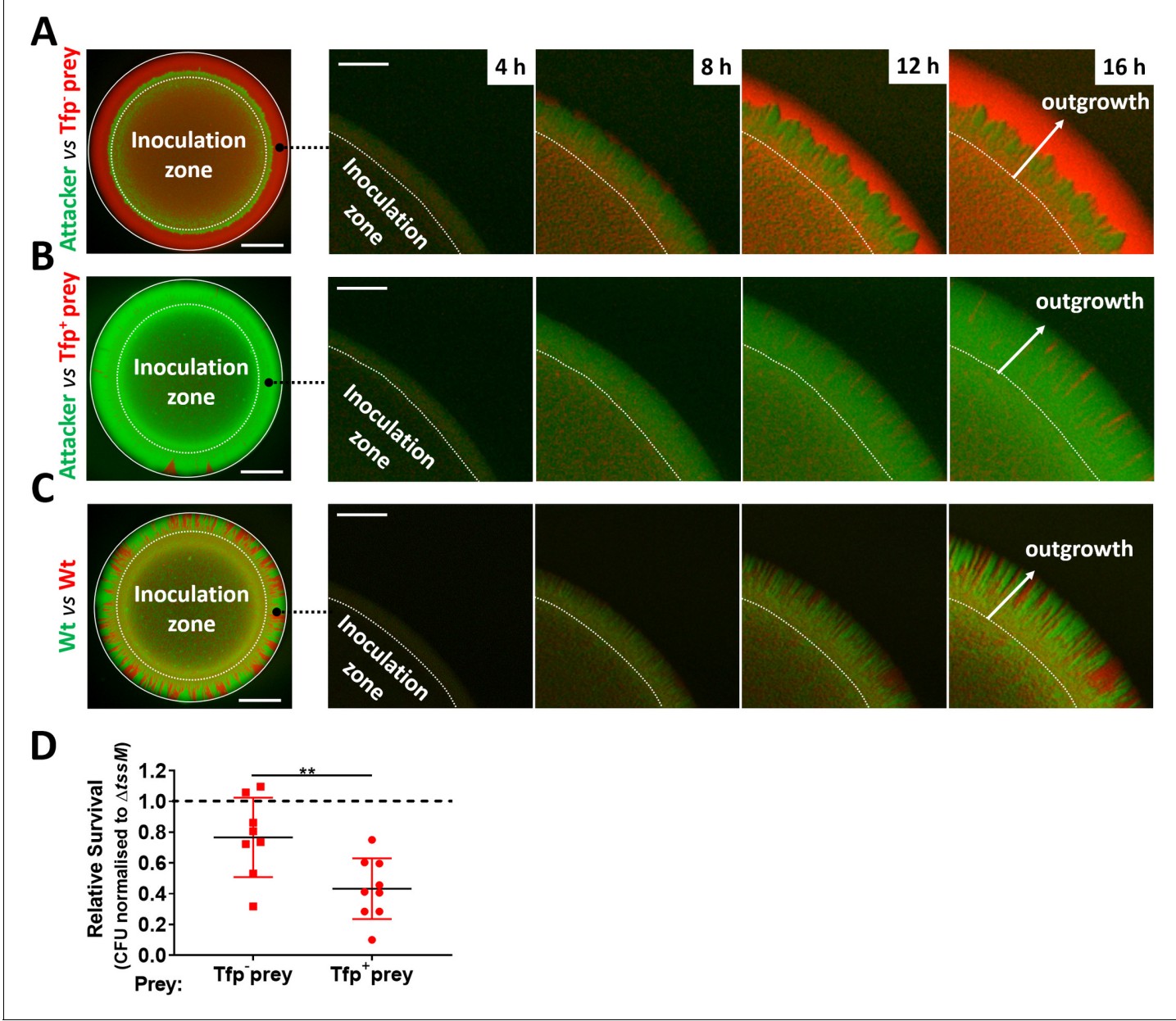

**Figure 7.** Attacker and prey piliation promotes T6SS killing. (A) Fluorescence microscopy images taken at specific times after inoculation of mixed (1:1 ratio) bacterial colonies. A T6SS-susceptible, non-piliated prey strain (346TΔ*nte/i3-5*Δ*pilE1/2_sfCherry*, red) migrates to the expanding edge of the colony over time, segregating from the T6SS+ attacker strain (*N. cinerea* 346T_*gfp*, green) and dominating the expanding population. See also *Figure 7—video 1*, (B) The same susceptible prey strain but expressing pili does not segregate, and after 24 hr is outcompeted by the piliated T6SS+ attacker. See also *Figure 7—video 2*. (C) The non-T6SS-susceptible, piliated prey strain (346T_*sfCherry*, red) and piliated attacker strain (346T_*sfGfp*, green) do not segregate, but due to immunity against T6SS attack, no dominance is observed. Images of colonies are representative of three independent experiments. See also *Figure 7—video 3*. Scale bar, 500 μm. Expanding colony edge images are stills at indicated times from time-lapse imaging performed on one occasion. Scale bar 100 μm. Flow cytometry data are presented in *Figure 7—figure supplement 2*. (D) The influence of piliation on T6SS killing. Recovery of non-piliated and piliated prey strains after 24 hr co-culture with *N. cinerea* 346T (Wt) and a *tssM*-deficient mutant (Δ*tssM*) at ratio of approx. 10:1, attacker:prey. Relative survival is defined as the fold change in recovery of prey following incubation with wild-type attacker *N. cinerea* compared to *N. cinerea* Δ*tssM*. Data shown are the mean ± SD of three independent experiments: **p < 0.01 using unpaired two-tailed Student's t-test for pairwise comparison. See also *Figure 7—figure supplement 3*.

The online version of this article includes the following video, source data, and figure supplement(s) for figure 7:

**Source data 1.** Survival of non-piliated and piliated prey strains after 24 hr co-culture with *N. cinerea* 346T and a *tssM*-deficient mutant.

**Figure supplement 1.** *N. cinerea* 346TΔ*nte/i3-5* prey is susceptible to T6SS-killing by wild-type *N. cinerea* 346T and fluorophore expressing mutants have comparable growth.

*Figure 7 continued on next page*

*Figure 7 continued*

**Figure supplement 2.** Flow cytometry analysis of relative proportion of attacker and prey strains.

**Figure supplement 3.** CFU and % survival data for Tfp+/- strains.

**Figure 7—video 1.** Growing edge of colonies with a piliated attacker *N. cinerea* 346T_*gfp*, (green) and non-piliated prey 346TΔ*nte*/*i3-5*Δ*pilE1*/*2_sfCherry* (red).

https://elifesciences.org/articles/63755#fig7video1

**Figure 7—video 2.** Growing edge of colonies with a piliated attacker *N. cinerea* 346T_*gfp*, (green) and piliated prey 346TΔ*nte*/*i3-5_sfCherry* (red).

https://elifesciences.org/articles/63755#fig7video2

**Figure 7—video 3.** Growing edge of colonies with two wild-type strains.

https://elifesciences.org/articles/63755#fig7video3

*2013*; *Stolle et al., 2021*). To address this, we compared the survival of piliated and non-piliated prey, in presence of Tfp+ or Tfp- attacker strain. Where pili are expressed on both or neither strain, segregation should not occur, enabling comparative analysis of the impact of Tfp on prey survival, independent of segregation. Competition assays to assess prey survival revealed that increased prey survival is only observed when the prey is non-piliated, but not when the attacker is non-piliated. Moreover, prey survival was equivalent when attacker and prey either both have, or both lack Tfp (*Figure 8A* and *Figure 8—figure supplement 1*). These data support the idea that the enhanced prey survival is due to segregation of the non-piliated prey from the piliated attacker, allowing the prey to achieve a favourable position for outgrowth at the edge of the colony.

We also used fluorescently labelled piliated and non-piliated attacker and prey to observe the same strains in mixed colonies as previously. Given that Tfp heterogeneity impacts segregation within a colony (*Oldewurtel et al., 2015*; *Pönisch et al., 2018*; *Zöllner et al., 2017*), when one of the two strains lacks pili, the non-piliated strain segregates and outgrowth is clearly visible at the edge of the colony (*Figure 8B* and *Figure 7A*). In the case where the non-piliated attacker segregates and dominates the colony edge, this appears to prevent any expansion of the prey, consistent with the lack of enhanced prey survival observed in competition assays. Interestingly, comparison of colonies where both attacker and prey are piliated with colonies where neither strain expresses Tfp revealed differences in prey expansion at the edge of the colony. In colonies with both strains lacking Tfp, we observe higher abundance of expanding sectors of the emerging prey population compared to colonies where attacker and prey cells are piliated (*Figure 8B* and *Figure 8—figure supplement 2*). One possible explanation is that Tfp interactions at the expanding edge bring adjacent attacker and prey cells into close proximity and thus result in a more effective reduction in the prey compared to when neither is piliated. Although competition assays did not reveal any difference in levels of prey survival when neither or both attacker and prey are piliated, this could be due to methodological limitations which mean that this very local effect is not detected at the population level. Further work is therefore necessary to explore the contribution of Tfp to T6SS-mediated attack beyond the impact on spatial reorganisation within a colony. Overall, data presented here confirm that Tfp influence the outcome of T6SS-mediated antagonism.

## Discussion

Here, we identified a T6SS in a commensal *Neisseria* spp. which can kill T6SS-deficient *N. cinerea* isolates and the related pathogens, *N. meningitidis*, with which it shares an ecological niche (*Knapp and Hook, 1988*), and *N. gonorrhoeae*. Of note, the *N. cinerea* T6SS is encoded on a large plasmid, with structural genes for the single T6SS apparatus clustered in one locus, similar to other T6SSs (*Anderson et al., 2017*; *Liaw et al., 2019*; *Sana et al., 2016*). To date, plasmid encoded T6SSs have only been described in *Campylobacter* species (*Marasini and Fakhr, 2016*), with this plasmid T6SS mobilised via conjugation (*Marasini et al., 2020*). Although other small plasmids have been reported in *N. cinerea* (*Knapp et al., 1984*; *Roberts, 1989*) and *N. cinerea* can be a recipient of *N. gonorrhoeae* plasmids (*Genco et al., 1984*), it is not yet known whether T6SS plasmids are widespread among *Neisseria*, or whether the plasmid can be mobilised by conjugation or transformation. Interestingly, in *Acinetobacter baylyi*, T6SS-induced prey cell lysis contributes to acquisition of plasmids from target cells (*Ringel et al., 2017*). Therefore, it will be interesting to see whether other *Neisseria* species with T6SS genes (*Marri et al., 2010*) harbour T6SS-expressing plasmids.

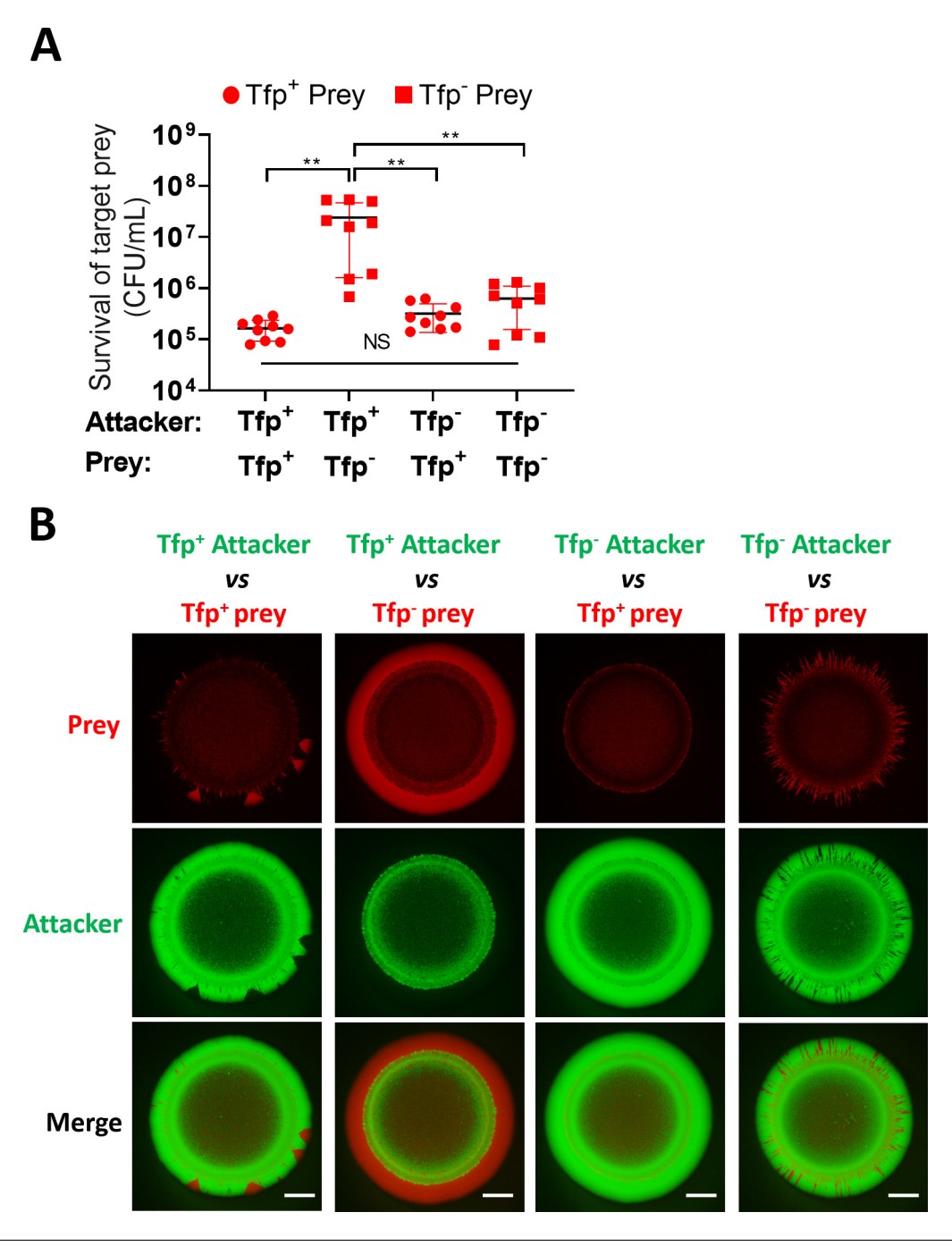

**Figure 8.** Tfp loss influences prey survival. (**A**) Role of Tfp in the attacker and prey population during competition. Recovery of non-piliated (346TΔ*nte/i3-5ΔpilE1/2_sfCherry*) and piliated prey (346TΔ*nte/i3-5_sfCherry*) strains after 24 hr co-culture with piliated *N. cinerea* 346T (346T_*sfGfp*) and non-piliated attacker (346TΔ*pilE1/2_sfGfp*) strains at ratio of approx. 10:1, attacker:prey. Data shown are the mean ± SD of three independent experiments: NS, not significant, **p<0.001 using one-way ANOVA test for multiple comparison. See also *Figure 8—figure supplement 1*. (**B**) Fluorescence microscopy images taken at 24 hr after inoculation of mixed (1:1 ratio) bacterial colonies. A T6SS-susceptible, piliated prey strain (Tfp+ prey, 346TΔ*nte/i3-5_sfCherry*, red) does not segregate, and after 24 hr is outcompeted by the piliated T6SS+ attacker (Tfp+ attacker, 346T_*sfGfp*, green). The same prey, but non-piliated (Tfp- prey, 346TΔ*nte/i3-5ΔpilE1/2_sfCherry*, red), segregates from the piliated T6SS+attacker strain (Tfp+ attacker, 346T_*sfGfp*, green) and dominates the edge of the colony. When the prey is piliated (Tfp+ prey, 346TΔ*nte/i3-5_sfCherry*, red) and attacker is non-piliated (Tfp- attacker, 346TΔ*pilE1/2_sfGfp*, green), the non-

*Figure 8 continued on next page*

*Figure 8 continued*

piliated attacker population segregates to the edge and dominates the outer region of the colony. In a mixed population with a non-piliated prey (Tfp- prey, 346TΔ*nte/i3-5ΔpilE1/2_sfCherry*, red) and a non-piliated attacker (Tfp- attacker, 346TΔ*pilE1/2_sfGfp*, green), the prey does not segregate from attacker and attacker and prey form expanding sectors in the region of outgrowth at the colony edge. Images of colonies are representative of three independent experiments. Scale bar, 500 μm. See also *Figure 8—figure supplement 2*.
The online version of this article includes the following source data and figure supplement(s) for figure 8:

**Source data 1.** Survival of non-piliated and piliated prey strains after 24 hr co-culture with piliated *N. cinerea* 346T and non-piliated attacker strains.
**Figure supplement 1.** % survival data for prey strains +/-Tfp.
**Figure supplement 2.** Fluorescence microscopy images of colonies of piliated (Tfp+/Tfp+) or non-piliated (Tfp-/Tfp-) attacker and prey strains.

---

In total six genes encoding putative effectors were identified based on their proximity to the T6SS locus, their pairwise arrangement with genes encoding proteins with homology to immunity proteins and the presence of conserved domains such as PAAR and Rhs domains in the predicted proteins (*Alcoforado Diniz et al., 2015*). Our bioinformatic predictions suggest Nte3, 4, 5, and 6 may be cargo effectors while Nte1 and Nte2 are more typical of specialised or evolved effectors with integral PAAR domains at their N-termini (*Durand et al., 2014*). Of note *nte5* encodes a protein with a shorter Rhs domain compared to the other five putative effectors (319AA compared to 695-735AA), raising the possibility that this may represent an orphan Rhs-CT, as described in other genomes (*Kirchberger et al., 2017*). Based on previous work, Hcp, or VgrG could be responsible for delivery of effectors encoded nearby in the T6SS locus (*Hachani et al., 2014*), with or without the involvement of the adjacent DUF2169 family protein (*Figure 1*). As *nte/nti6* are not part of the T6SS locus and are encoded elsewhere on the plasmid, Nte/Nti6 may not be associated with the T6SS. Thus, although our bioinformatic analysis and protein expression in *E. coli* suggests Nte1-6 as effectors, additional experimentation will be needed to further confirm their contributions to T6SS activity and killing, and to demonstrate direct secretion *via* the T6SS.

Examination of *N. cinerea* T6SS activity revealed several interesting features. Microscopy demonstrated that T6SS attack (tit-for-tat) is not required to provoke firing of the system. Instead, the T6SS appears to be constitutively active in *N. cinerea* (*Figure 2*). Furthermore, the system is capable of inducing lysis of prey bacteria (*Figure 3*). The consequences of T6SS attack are determined by the repertoire and activities of effectors, and their site of delivery. Many different effector activities have been proposed including lipases, peptidoglycan hydrolases, metalloproteases, and nucleases (*Lewis et al., 2019*). Effector activities can result in target cell lysis to varying degrees (*Ringel et al., 2017*; *Smith et al., 2020*). Of the six Ntes we identified, lysis could be mediated by Nte1 which harbours a putative phospholipase domain in the C-terminus. Alternatively, a combination of effectors might be needed to elicit prey lysis.

Polysaccharide capsules are largely thought to provide bacteria with a strategy for evading host immune killing (*Lewis and Ram, 2014*). Here, we found that the meningococcal capsule has an alternative role in defence against other bacteria. Meningococcal strains lacking a capsule were at a significant disadvantage in the face of a T6SS-expressing competitor implicating this surface polysaccharide in protection against T6SS assault. Similar findings have been reported for other bacteria; for example, the extracellular polysaccharide of *V. cholerae* and the colanic acid capsule of *E. coli* confer defence against T6SS attack (*Hersch et al., 2020*; *Toska et al., 2018*). One potential mechanism is that the capsule sterically impairs the ability of the T6SS to penetrate the target cell membrane, and/or inhibits access of T6SS effectors to their cellular targets. Interestingly, recent genetic evidence indicates that some commensal *Neisseria* species also have capacity to produce polysaccharide capsules (*Clemence et al., 2018*), which might also confer a survival advantage in mixed populations that include strains expressing T6SS.

Most bacteria exist within complex polymicrobial communities in which the spatial and temporal dynamics of proliferation and death have a major effect on their fitness and survival (*Nadell et al., 2016*). While structured complex microbial societies can benefit all their members (*Gabrilska and Rumbaugh, 2015*; *Wolcott et al., 2013*), antagonistic neighbours, especially those deploying contact-dependent killing mechanisms, can disrupt communities. Although T6SS-mediated killing can be

advantageous to a producing strain during bacterial competition, this requires intimate association with its prey (*MacIntyre et al., 2010*; *Russell et al., 2014*). Thus, one way for susceptible bacteria to evade T6SS killing is to avoid direct contact with attacking cells (*Borenstein et al., 2015*; *Smith et al., 2020*). In *Neisseria*, the Tfp is a key mediator of interbacterial and interspecies interactions (*Custodio et al., 2020*; *Higashi et al., 2011*) and pilus-mediated interactions influence the spatial structure of a growing community (*Oldewurtel et al., 2015*; *Zöllner et al., 2017*). In *N. gonorrhoeae,* non-piliated bacteria segregate to the expanding front of the colony and Tfp-mediated spatial reorganisation can allow bacteria to avoid external stresses or strains competing for resources (*Oldewurtel et al., 2015*; *Zöllner et al., 2017*). We predicted that this would be especially relevant in the context of T6SS-mediated antagonism. For example, physical exclusion driven by Tfp-loss or modification could be an effective strategy to evade and survive an antagonistic interaction, while pilus-mediated interactions might be less favourable for a susceptible prey. Importantly, Tfp loss may occur naturally in a polymicrobial environment and is an established phenomenon in pathogenic *Neisseria* (*Hagblom et al., 1985*; *Helm and Seifert, 2010*). Our results demonstrate that within a bacterial community of attacker and prey cells with or without pili, the sorting of the non-piliated prey to the colony edge results in enhanced survival, likely through segregation of the prey from the attacker. Reduction in prey survival was equivalent whether both or neither the attacker and prey express Tfp, demonstrating that Tfp are not required for T6SS activity. However, observation of the prey in mixed colonies where both strains are piliated compared to when neither strain expresses Tfp suggests a possible localised contribution of Tfp. It will be interesting to further explore the contribution of Tfp to T6SS activity at the single-cell level, to ascertain their localised impact and explore for example how this affected by pilus retraction or pilin sequence variation. It is noteworthy that many bacteria (e.g. *Pseudomonas aeruginosa*, *Vibrio cholerae*, *Acinetobacter baumannii*, enteropathogenic *E. coli*) that employ T6SSs for inter-bacterial competition also express Tfp. Therefore, our findings are of broad relevance for the impact of contact-dependent killing, and further emphasise how precise spatial relationships can have profound effects on how antagonistic and mutualistic factors combine to influence the development of microbial communities.

## Materials and methods

### Bacterial strains and growth
Bacterial strains used in this study are shown in Key Resources Table (Appendix). *Neisseria* spp. were grown on Brain Heart Infusion (BHI, Oxoid) agar with 5% defibrinated horse blood or in BHI broth at 37°C with 5% $CO_2$ or GC-medium supplemented with 1.5% base agar (w/v) and 1% Vitox (v/v; Oxoid). GW-medium (*Wade and Graver, 2007*) was used for *N. cinerea* microscopy experiments. *E. coli* was grown on LB (Lennox Broth base, Invitrogen) agar or in liquid LB at 37°C with shaking. Antibiotics were added at the following concentrations: for *E. coli*, carbenicillin (carb) 100 µg/ml, kanamycin (kan) 50 µg/ml, and chloramphenicol (cm) 20 µg/ml; for *Neisseria* spp. kan 75 µg/ml, spectinomycin (spec) 65 µg/ml, erythromycin (ery) 15 µg/ml, and polymyxin B (pmB) 10 µg/ml.

### DNA isolation and whole-genome sequencing (WGS)
Genomic DNA was extracted using the Wizard Genomic Kit (Promega), and sequenced by PacBio (Earlham Institute, Norwich) using single-molecule real-time (SMRT) technology; reads were assembled de novo with HGAP3 (*Chin et al., 2013*).

### Bioinformatic analysis of putative T6SS genes
All ORFs on the *N. cinerea* 346T plasmid were analysed manually using NCBI BLASTp against non-redundant protein databases at NCBI using default search parameters to confirm the presence of T6SS-associated conserved domains. The PAAR-like domain and Rhs domain from *N. cinerea* 346T Nte1 plus VgrG amino acid sequences from *N. cinerea* 346T T6SS locus were used as query sequences in BLASTp analysis using the PubMLST BLAST tool. The default parameter of word size (length of the initial identical match that is required before extending a hit) of 11 was used for all searches. Output of 10 hits per isolate was selected to enable identification of multiple homologues within a genome. BLAST results were then subjected to further manual refinement by filtering the hits obtained using a cut-off of at least 20% homology to the query sequence and 20% coverage. The

FASTA nucleotide sequence of the hits (including 100 bp flanking sequence) were extracted from the PubMLST database and mapped onto the 346T PacBio genome using SnapGene. Each of the ORFs mapped on the genome were further analysed by NCBI BLASTx against non-redundant protein databases at NCBI using default search parameters to confirm the presence of T6SS-associated conserved domains. T6SS-effector prediction software tools (*Li et al., 2015*) were also used to identify putative effectors.

## Construction of *N. cinerea* mutants

Primers used in this study are listed in key resources table (Appendix). Target genes were replaced with antibiotic cassettes as previously (*Wörmann et al., 2016*). Constructs were assembled into pUC19 by Gibson Assembly (New England Biolabs), and hosted in *Escherichia coli* DH5α. Plasmids were linearised with *Sca*I, and gel extracted, relevant linearised fragments used to transform *N. cinerea*; transformants were checked by PCR and sequencing. Complementation or chromosomal insertion of genes encoding fluorophores was achieved using pNCC1-Spec, a spectinomycin-resistant derivative of pNCC1 (*Wörmann et al., 2016*). For visualisation of T6SS-sheaths, *sfgfp* was cloned in-frame with *tssB* and a short linker (encoding 3×Ala 3×Gly) by Gibson Assembly (New England Biolabs) into pNCC1-Spec to allow IPTG-inducible expression of TssB-sfGFP. PCR was performed using Herculase II (Agilent) or Q5 High-fidelity DNA Polymerase (New England Biolabs).

## Analysis of effector/immunity activity in *E. coli*

Putative effector coding sequences with or without cognate immunity gene were amplified by PCR from *N. cinerea* 346T gDNA and either assembled by Gibson Assembly (NEB) into pBAD33 or, for Nte1 with or without addition of the PelB signal sequence, cloned in to pBAD33 using XbaI / SphI restriction enzyme sites. All forward primers also included the *E. coli* ribosomal binding site (RBS: AAGAAGG) upstream of the start codon. Plasmids were transformed into *E. coli* DH5α and verified by sequencing (Source Bioscience). For assessment of toxicity, strains with recombinant or empty pBAD33 plasmids were grown overnight in LB supplemented with 0.8% glucose (w/v), then diluted to an $OD_{600}$ of 0.1 and incubated for 1 hr at 180 rpm and 37°C; bacteria were pelleted and resuspended in LB with arabinose (0.8% w/v) to induce expression and incubated at 37°C, 180 rpm for a further 4 hr. The $OD_{600}$ and CFU/ml of cultures were determined; aliquots were diluted and plated to media containing 0.8% glucose at relevant time points up to 5 hr.

## Hcp protein expression, purification, and antibody generation

Codon optimised *hcp* was synthesised with a sequence encoding an N-terminal 6x His Tag and a 3C protease cleavage site, and flanked by *Nco*I and *Xho*I restriction sites (ThermoFisher). The fragment was ligated into *Nco*I and *Xho*I sites in pET28a (Novagen) using QuickStick T4 DNA Ligase (Bioline) and transformed into *E. coli* B834. Bacteria were grown at 37°C, 150 rpm to an $OD_{600}$ of 1.0, and expression of 6xHis-3C-Hcp was induced with 1 mM IPTG for 24 hr at 16°C. Cells were resuspended in Buffer A (50 mM Tris-HCl buffer pH 7.5, 10 mM Imidazole, 500 mM NaCl, 1 mM DTT) containing protease inhibitors, 1 mg/mL lysozyme and 100 μg/mL DNase then subsequently homogenised with an EmulsiFlex-C5 (Avestin). Lysed cells were ultracentrifuged, and the cleared supernatant loaded onto a Ni Sepharose 6 Fast Flow His Trap column (GE Healthcare) equilibrated with Buffer A. The column was washed with Buffer A, then Buffer B (50 mM Tris-HCl buffer pH 7.5, 35 mM Imidazole, 500 mM NaCl, 1 mM DTT) before elution with 10 mL of Buffer C (50 mM Tris-HCl buffer pH 7.5, 300 mM Imidazole, 150 mM NaCl, 1 mM DTT). The eluate was incubated with the HRV-3C protease (Sigma) then applied to a Ni Sepharose column. The eluate containing protease and cleaved protein was concentrated using Amicon Ultra 10,000 MWCO (Millipore), then passed through a Superdex-200 column (GE Healthcare, Buckinghamshire, UK). Fractions were analysed by SDS-PAGE and Coomassie blue staining, and those with Hcp pooled and used to generate polyclonal antibodies (EuroGentec).

## Hcp secretion assay

Bacteria were grown in BHI broth for 4–5 hr then harvested and lysed in an equal volume of SDS-PAGE lysis buffer (500 mM Tris-HCl [pH 6.8], 5% SDS, 15% glycerol, 0.5% bromophenol blue containing 100 mM β-mercaptoethanol); supernatants were filtered (0.22 μm pore, Millipore) and

proteins precipitated with 20% (v/v) trichloroacetic acid. Hcp was detected by Western blot with anti-Hcp (1:10,000 dilution) and goat anti-rabbit IgG–HRP (1:5000, sc-2004; Santa Cruz). Anti-RecA (1:5000 dilution, ab63797; Abcam) followed by goat anti-rabbit IgG–HRP and detection with ECL detection Reagent (GE Healthcare) or Coomassie blue staining were used as loading controls.

### Live cell imaging of T6SS activity

Bacteria were grown overnight on BHI agar, resuspended in PBS and 20 µl spotted onto fresh BHI agar containing 1 mM IPTG and incubated for 4 hr at 37°C. After incubation, 500 µl of $10^9$ CFU/mL bacterial suspension of attacker was mixed with the prey strain at a 1:1 ratio. Cells were harvested by centrifugation for 3 min at 6000 rpm, resuspended in 100 µL of PBS or GW media and 2 µl spotted on 1% agarose pads (for T6SS dynamics) or GW media with 0.1 mM IPTG and 0.5 µM SYTOXBlue (Thermo Fisher Scientific) for assessment of prey permeability. Fluorescence microscopy image sequences were acquired within 20–30 min of sample preparation with an inverted Zeiss 880 Airyscan microscope equipped with Plan-Apochromat 63×/1.4-NA oil lens and fitted with a climate chamber mounted around the objective to perform the imaging at 37°C with 5% $CO_2$. Automated images were collected at 1 s, 10 s or 1 min intervals and processed with Fiji (*Schindelin et al., 2012*). Background noise was reduced using the 'Despeckle' filter. The XY drift was corrected using StackReg with 'Rigid Body' transformation (*Thévenaz et al., 1998*). Experiments and imaging were performed on at least two independent occasions.

### Quantitative competition assays

Strains grown overnight on BHI agar were resuspended in PBS and diluted to $10^9$ CFU/mL based on OD quantification, mixed at an approximate ratio of ~10:1 for *N. cinerea/ N. cinerea* and *N. cinerea/ N. gonorrhoeae*, or ~100:1 for *N. cinerea/ N. meningitidis* (actual CFU are indicated in source data files where available), then 20 µl spotted onto BHI agar in triplicate and incubated at 37°C with 5% $CO_2$. At specific time-points, entire spots were harvested and resuspended in 1 mL of PBS. The cellular suspension was then serially diluted in PBS and aliquots spotted onto selective media. Colonies were counted after ~16 hr incubation at 37°C with 5% $CO_2$. Experiments were performed on at least three independent occasions. For different prey analysis, relative survival was defined as the fold change in recovery of prey following incubation with wild-type attacker *N. cinerea* compared to a T6SS-deficient *N. cinerea*.

### Competition assays assessed by fluorescence microscopy and flow cytometry

Bacteria were grown overnight on BHI, resuspended in PBS and diluted to $10^9$ CFU/mL. 100 µl of each suspension (attacker/prey) were mixed thoroughly (*i.e.*, a 1:1 ratio) and 1 µl spotted in duplicate onto GC-medium supplemented with 0.5% base agar (w/v) and 1% Vitox (v/v; Oxoid). Plates were incubated for 24 hr at 37°C, 5% $CO_2$. For flow cytometry analysis, the remaining input suspension was then centrifuged for 3 mins at 6000 rpm then pellets resuspended in 500 µL of 4% paraformaldehyde and fixed for 20 min at room temperature. Following centrifugation, the fixed bacteria were then resuspended in 250 µL PBS and stored at 4°C for 24 hr prior to analysis. At various time points, expanding colonies were imaged using a M125C stereo microscope equipped with a DFC310FX digital camera (Leica Microsystems), and images processed with Fiji. Images were imported using 'Image Sequence' and corrected with StackReg as described above. At 24 hr, colonies were harvested, fixed with 4% PFA for 20 min then washed with PBS. Samples were analysed using a Cytoflex LX (Beckman Coulter), and at least $10^4$ events recorded. Fluorescence, forward and side scatter data were collected to distinguish between debris and bacteria. Results were analysed by calculating the number of events positive for either GFP or Cherry signal in FlowJo v10 software (Becton Dickinson Company). The negative population (non-fluorescent cells) was established using 346T Wt, the GFP+ population was determined using *N. cinerea* 346T Wt_sfGFP, and the Cherry+ population using *N. cinerea* 346T Wt_sfCherry. Quadrants were set to delineate the GFP+, Cherry+, GFP+Cherry+ and the percentage of cells representing each population within the different competition spots was recorded. Flow cytometry analysis was performed on two independent occasions. Stereo microscopy analysis was performed on three independent occasions with technical duplicates each time.

## Statistical analyses

Graphpad Prism7 software (San Diego, CA) was used for statistical analysis. We used One-way/two-way ANOVA with Tukey post hoc testing for multiple comparisons and unpaired two-tailed Student's t-test for pairwise comparisons. In all cases, $p < 0.05$ was considered statistically significant.

# Acknowledgements

We thank members of the Foster group (Oxford) especially Daniel Unterweger (now at the University of Kiel) for advice and assistance with microscopy as well as Alan Wainman of the SWDSP Bioimaging facility. We are grateful to M Basler (Basel) for valuable advice, and to Meningitis Now for funding. Work in CMT's lab is supported by a Wellcome Trust Investigator award (102908/Z/13/Z).

# Additional information

## Competing interests

Rachel M Exley: Previous Member of the Scientific Advisory Panel of Meningitis Research Foundation (until 2021). The other authors declare that no competing interests exist.

## Funding

| Funder | Grant reference number | Author |
| --- | --- | --- |
| Wellcome Trust | 102908/Z/13/Z | Christoph M Tang |
| Meningitis Now | | Christoph M Tang<br>Rachel M Exley |

The funders had no role in study design, data collection and interpretation, or the decision to submit the work for publication.

## Author contributions

Rafael Custodio, Conceptualization, Formal analysis, Validation, Investigation, Visualization, Methodology, Writing - original draft, Writing - review and editing; Rhian M Ford, Cara J Ellison, Formal analysis, Validation, Investigation, Visualization, Writing - original draft; Guangyu Liu, Resources; Gerda Mickute, Formal analysis, Validation, Investigation, Visualization; Christoph M Tang, Rachel M Exley, Conceptualization, Supervision, Funding acquisition, Validation, Writing - original draft, Project administration, Writing - review and editing

## Author ORCIDs

Rafael Custodio (iD) https://orcid.org/0000-0002-7561-5515
Gerda Mickute (iD) https://orcid.org/0000-0002-2477-7565
Christoph M Tang (iD) http://orcid.org/0000-0001-8366-3245
Rachel M Exley (iD) https://orcid.org/0000-0001-9120-5586

## Decision letter and Author response

Decision letter https://doi.org/10.7554/eLife.63755.sa1
Author response https://doi.org/10.7554/eLife.63755.sa2

# Additional files

## Supplementary files

- Supplementary file 1. Table of Putative T6SS core components in *N. cinerea* 346T.

- Transparent reporting form

## Data availability

All data generated or analysed in this study are included in the manuscript and supporting files. Source data files have been provided for Figures 1, 2, 5, 6, 7 and 8 and for Figure Supplements 2, 5 and 7. Whole genome sequence data has been deposited in Dryad (doi: https://doi.org/10.5061/dryad.3ffbg79gx).

The following dataset was generated:

| Author(s) | Year | Dataset title | Dataset URL | Database and Identifier |
|---|---|---|---|---|
| Custodio R | 2020 | *Neisseria cinerea* 346T whole genome sequence | http://dx.doi.org/10.5061/dryad.3ffbg79gx | Dryad Digital Repository, 10.5061/dryad.3ffbg79gx |

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

# Appendix 1

**Appendix 1—key resources table**

| Reagent type (species) or resource | Designation | Source or reference | Identifiers | Additional information |
|---|---|---|---|---|
| Antibody | Goat polyclonal anti-rabbit IgG–HRP | Santa Cruz | sc-2004 | Target rabbit IgG antibodies WB (1:5000) |
| Antibody | Rabbit polyclonal anti-RecA | Abcam | ab63797 | Target bacterial RecA protein WB (1:5000) |
| Antibody | Rabbit polyclonal anti-Hcp sera | This paper | | Antibody raised to target full-length *N. cinerea* 346T Hcp protein WB (1:10000) |
| Chemical compound, drug | SYTOX Blue | Thermo Fisher Scientific | S34857 | SYTOX Blue is a high-affinity nucleic acid stain that does not penetrate uncompromised cell membranes |
| Software, algorithm | FiJi | *Schindelin et al., 2012* DOI:10.1038/nmeth.2019 | https://fiji.sc RRID:SCR_002285 | |
| Software, algorithm | Graphpad Prism7 | San Diego, CA | https://www.graphpad.com/ RRID:SCR_002798 | |
| Software, algorithm | FlowJo v10 | Becton Dickinson Company | https://www.flowjo.com/ RRID:SCR_008520 | |
| Strain, strain background (*Neisseria cinerea*) | CCUG346T (346) | *Bennett et al., 2012* DOI:10.1099/mic.0.056077-0 | | wild-type *N. cinerea* |
| Strain, strain background (*Neisseria cinerea*) | CCUG27178A (27178A) | *Bennett et al., 2012* DOI:10.1099/mic.0.056077-0 | | wild-type *N. cinerea* |
| Strain, strain background (*Neisseria cinerea*) | 346T_sfGFP | *Wörmann et al., 2016* DOI: 10.1099/mic.0.000248 | | 346T with chromosomally integrated *sfGfp*; Ery[R] |
| Strain, strain background (*Neisseria cinerea*) | 346T_sfGFPΔ*pilE*1/2 | *Wörmann et al., 2016* DOI: 10.1099/mic.0.000248 | | 346T *with pilE*1 and *pilE*2 deleted by insertion mutagenesis, and chromosomally integrated *sfGfp*; Ery[R] and Kan[R] |
| Strain, strain background (*Neisseria cinerea*) | 346T_sfCherry | This paper | | 346T with chromosomally integrated *sfCherry*; Ery[R] |

*Continued on next page*

*Appendix 1—key resources table continued*

| Reagent type (species) or resource | Designation | Source or reference | Identifiers | Additional information |
|---|---|---|---|---|
| Strain, strain background (*Neisseria cinerea*) | 27178A_sfCherry | This paper | | 27178 with chromosomally integrated *sfCherry*; Spec[R] |
| Strain, strain background (*Neisseria cinerea*) | 346TΔT6SS | This paper | | 346T with insertion-deletion of *tssC – vgrG* region; Ery[R] |
| Strain, strain background (*Neisseria cinerea*) | 346TΔ*tssB* | This paper | | 346T with insertion-deletion of *tssB*; Ery[R] |
| Strain, strain background (*Neisseria cinerea*) | 346TΔ*tssB::tssBsfGFP* | This paper | | 346T with insertion-deletion of native *tssB* and ectopic chromosomal insertion of *tssB-sfGFP* fusion; Spec[R] Ery[R] |
| Strain, strain background (*Neisseria cinerea*) | 346TΔ*tssM* | This paper | | 346T with insertion-deletion of *tssM*; Tet[R] |
| Strain, strain background (*Neisseria cinerea*) | 346TΔ*tssB*Δ*tssM::tssB-sfGFP* | This paper | | 346T with insertion-deletion of native *tssB* and *tssM* and ectopic chromosomal insertion of *tssB-sfGFP* fusion; Spec[R] Ery[R] Tet[R] |
| Strain, strain background (*Neisseria cinerea*) | 346TΔ*nte3*Δ*nte4*Δ*nte5* | This paper | immunity genes; Ery[R] | deletion mutagenesis, *nte3-nte5* locus deletion including respective immunity genes; Ery[R] |
| Strain, strain background (*Neisseria cinerea*) | 346TΔ*nte/i3-5_sfCherry* | This paper | | 346T with insertion-deletion of *nte/i3-5* region and ectopic chromosomal insertion of *sfCherry*; Spec[R] Ery[R] |
| Strain, strain background (*Neisseria cinerea*) | 346TΔ*nte6* | This paper | | deletion mutagenesis, *nte6* deficient; Spec[R] |
| Strain, strain background (*Neisseria cinerea*) | 346TΔ*nte3*Δ*nte4*Δ*nte5*Δ*nte6* | This paper | | deletion mutagenesis, *nte3-nte5* locus deletion including respective immunity genes plus *nte6* deletion; Ery[R] SpecR |

*Continued on next page*

*Appendix 1—key resources table continued*

| Reagent type (species) or resource | Designation | Source or reference | Identifiers | Additional information |
|---|---|---|---|---|
| Strain, strain background (*Neisseria cinerea*) | 346TΔ*nte/i3-5*Δ*pilE1/2_sfCherry* | This paper | | 346T with insertion-deletion of *nte/i3-5* region; ectopic chromosomal insertion of *sfCherry*; insertion-deletion of *pilE1* and *pilE2*; kan^R, Spec^R Ery^R |
| Strain, strain background (*Neisseria meningitidis*) | 8013 | *Rusniok et al., 2009* DOI: 10.1186/gb-2009-10-10-r110 | | *N. meningitidis* wild-type |
| Strain, strain background (*Neisseria meningitidis*) | MC58 | *Tettelin et al., 2000* DOI: 10.1126/science.287.5459.1809. | | *N. meningitidis* wild-type |
| Strain, strain background (*Neisseria meningitidis*) | S3 | *Uria et al., 2008* DOI: 10.1084/jem.20072577 | | *N. meningitidis* wild-type |
| Strain, strain background (*Neisseria meningitidis*) | MC58Δ*siaD* | *Virji et al., 1995* DOI:10.1111/j.1365-2958.1995.mmi_18040741.x | | deletion mutagenesis, NEIS0051; Kan^R |
| Strain, strain background (*Neisseria gonorrhoeae*) | FA1090 pGCC4 | *Mehr and Seifert, 1997* DOI:10.1046/j.1365-2958.1997.2971660.x | | FA1090 with chromosomally integrated plasmid pGCC4; Ery^R |
| Strain, strain background (*Escherichia coli*) | Dh5α | Lab collection | | DH5α is an *E. coli* strain used for general cloning applications. |
| Strain, strain background (*Escherichia coli*) | Dh5α pNCC1-Spec | This paper | | Dh5α with pNCC1Spec^R plasmid |
| Strain, strain background (*Escherichia coli*) | Dh5α pNCC1-Spec-sfGFP | This paper | | Dh5α with pNCC1-Spec with *sfGFP* insert; |
| Strain, strain background (*Escherichia coli*) | Dh5α pNCC101-Spec-sfCherry | Lab collection | | DH5α with plasmid pNCC101+*sfCherry* insert. Spec^R |
| Strain, strain background (*Escherichia coli*) | Dh5α pUC19 | Lab collection | pUC19 vector RRID:Addgene_50005 | *E. coli* DH5α strain harbouring pUC19 for general cloning applications. |
| Strain, strain background (*Escherichia coli*) | Dh5α pUC19::Δ*tssB* | This paper | | DH5α with pUC19::Δ*tssB* deletion construct; Carb^R Ery^R |

*Continued on next page*

*Appendix 1—key resources table continued*

| Reagent type (species) or resource | Designation | Source or reference | Identifiers | Additional information |
|---|---|---|---|---|
| Strain, strain background (*Escherichia coli*) | Dh5α pUC19::Δ*tssM* | This paper | | DH5α with pUC19::Δ*tssM* deletion construct; Carb^R Tet^R |
| Strain, strain background (*Escherichia coli*) | Dh5α pUC19::ΔT6SS | This paper | | DH5α with pUC19::Δ*tssC-vgrG* locus deletion construct; Carb^R Ery^R |
| Strain, strain background (*Escherichia coli*) | Dh5αpUC19::Δ*nte3*Δ*nte4*Δ*nte5* | This paper | | DH5α with pUC19::Δ*nte3*Δ*nte4*Δ*nte5* region including respective immunity genes deletion construct; Carb^R Ery^R |
| Strain, strain background (*Escherichia coli*) | Dh5α pUC19:: Δ*nte6* | This paper | | *nte6* deletion construct; Carb^R Spec^R |
| Strain, strain background (*Escherichia coli*) | B834 pET28a | Lab collection | pET28a Novagen Cat. No. 69864–3 | B834 with pET28a IPTG-inducible expression vector, Kan^R |
| Strain, strain background (*Escherichia coli*) | Dh5α pET28a-His-3C-Hcp | This paper | | Dh5α with pET28a vector for IPTG inducible expression of Nc 346T Hcp with N-terminal cleavable HIS tag. Kan^R |
| Strain, strain background (*Escherichia coli*) | B834 pET28a-His-3C-Hcp | This paper | | B834 expression strain, with pET28a vector for IPTG inducible expression of Nc 346T Hcp with N-terminal cleavable HIS tag. Kan^R |
| Strain, strain background (*Escherichia coli*) | Dh5α pBAD33 | Lab collection | pBAD33 RRID:Addgene_36267 | Dh5α with pBAD33 vector for Arabinose-inducible expression, Cm^R |
| Strain, strain background (*Escherichia coli*) | Dh5α pBAD33::(ssPelB) Nte1-His | This paper | | Dh5α with pBAD33 encoding Nte1 with N-terminal PelB leader peptide and C-terminal his-tag under arabinose-inducible promoter control; Cm^R |
| Strain, strain background (*Escherichia coli*) | Dh5α pBAD33:: (ssPelB) Nte1+Nti1 | This paper | | Dh5α with pBAD33 encoding Nte1 with N-terminal PelB leader peptide and C-terminal his-tag plus Nti, under arabinose-inducible promoter control; Cm^R |

*Continued on next page*

*Appendix 1—key resources table continued*

| Reagent type (species) or resource | Designation | Source or reference | Identifiers | Additional information |
|---|---|---|---|---|
| Strain, strain background (*Escherichia coli*) | Dh5α pBAD33::Nte1-His | This paper | | Dh5α with pBAD33 encoding Nte1 with N-terminal his-tag under arabinose-inducible promoter control; $Cm^R$ |
| Strain, strain background (*Escherichia coli*) | Dh5α pBAD33::Nte2 | This paper | | Dh5α with pBAD33 encoding Nte2 under arabinose-inducible promoter control; $Cm^R$ |
| Strain, strain background (*Escherichia coli*) | Dh5α pBAD33::Nte2+Nti2 | This paper | | Dh5α with pBAD33 encoding Nte2+Nti2 under arabinose-inducible promoter control; $Cm^R$ |
| Strain, strain background (*Escherichia coli*) | Dh5α pBAD33::Nte3 | This paper | | Dh5α with pBAD33 encoding Nte3 under arabinose-inducible promoter control; $Cm^R$ |
| Strain, strain background (*Escherichia coli*) | Dh5α pBAD33::Nte3+Nti3 | This paper | | Dh5α with pBAD33 encoding Nte3+Nti3 under arabinose-inducible promoter control; $Cm^R$ |
| Strain, strain background (*Escherichia coli*) | Dh5α pBAD33::Nte4 | This paper | | Dh5α with pBAD33 encoding Nte4 under arabinose-inducible promoter control; $Cm^R$ |
| Strain, strain background (*Escherichia coli*) | Dh5α pBAD33::Nte4+Nti4 | This paper | | Dh5α with pBAD33 encoding Nte4+Nti4 under arabinose-inducible promoter control; $Cm^R$ |
| Strain, strain background (*Escherichia coli*) | Dh5α pBAD33::Nte5 | This paper | | Dh5α with pBAD33 encoding Nte5 under arabinose-inducible promoter control; $Cm^R$ |
| Strain, strain background (*Escherichia coli*) | Dh5α pBAD33::Nte5+Nti5 | This paper | | Dh5α with pBAD33 encoding Nte5+Nti5 under arabinose-inducible promoter control; $Cm^R$ |
| Strain, strain background (*Escherichia coli*) | Dh5α pBAD33::Nte6$^{R1300S}$ | This paper | | Dh5α with pBAD33 encoding Nte6$^{R1300S}$ under arabinose-inducible promoter control; $Cm^R$ |
| Strain, strain background (*Escherichia coli*) | Dh5α pBAD33::Nte6+Nti6 | This paper | | Dh5α with pBAD33 encoding Nte6+Nit6 under arabinose-inducible promoter control; $Cm^R$ |

*Continued on next page*

*Appendix 1—key resources table continued*

| Reagent type (species) or resource | Designation | Source or reference | Identifiers | Additional information |
|---|---|---|---|---|
| Sequence-based reagent | T6SSdel-1 | This paper | 5'-CGAAAAGTG CCACCTGACGTATGA CTGAAAAGCAATTAGATA TC | Deletion of *tssC-vgrG* locus |
| Sequence-based reagent | T6SSdel-2 | This paper | 5'-GTTAAATTTAAGGA TAAGAAACGTGGCAG | Deletion of *tssC-vgrG* locus |
| Sequence-based reagent | T6SSdel-3 | This paper | 5'-TTTCTTATCC TTAAATTTAACG ATCACTCATCATG | Deletion of *tssC-vgrG* locus |
| Sequence-based reagent | T6SSdel-4 | This paper | 5'-ACTCAAACATTTACTTAT TAAATAATTTATAGCTA TTGAAAAG | Deletion of *tssC-vgrG* locus |
| Sequence-based reagent | T6SSdel-5 | This paper | 5'-TTAATAAGTAAATGTTT GAGTTGCAGAACTTTAC | Deletion of tssC-vgrG locus |
| Sequence-based reagent | T6SSdel-6 | This paper | 5'-GATAATAATGGTTTC TTAGAC GTGCCGTTCCAA TAGGCCATAG | Deletion of *tssC-vgrG* locus |
| Sequence-based reagent | T6SSdel-conf-F | This paper | 5'-CCTAAAGCG GCTTCCAAAGACG | Confirmation of *tssC-vgrG* locus deletion |
| Sequence-based reagent | T6SSdel-conf-R | This paper | 5'-CCATGCCGG TAAAGGTCAGT | Confirmation of *tssC-vgrG* locus deletion |
| Sequence-based reagent | TssBdel-1 | This paper | 5'-GATCCTCTA GAGTCGACCTGCAGGCA TGCACTTACCCTGATC CACAAAGCC | Deletion of *tssB* |
| Sequence-based reagent | TssBdel-2 | This paper | 5'-ATTCAATGACCTTTAAA TGATAAAGTTGT | Deletion of *tssB* |
| Sequence-based reagent | TssBdel-3 | This paper | 5'-ACAACTTTTATCA TTTAAAG GTCATTGAATA TGAACGAGAA AAATATAAAACACAGTC | Deletion of *tssB* |
| Sequence-based reagent | TssBdel-4 | This paper | 5'-TTACTTATTA AATAATTTATAGCTATTGA AAAGAGATAAGAATTG | Deletion of *tssB* |
| Sequence-based reagent | TssBdel-5 | This paper | 5'-TATAAATTATTTAATAAG TAAG CTTCCAAAGACGAGCAG TAA | Deletion of *tssB* |
| Sequence-based reagent | TssBdel-6 | This paper | 5'-CAGGAAACA GCTATGACCATGATTACG CCTAAGTTGCGGGCAAC TTCTT | Deletion of *tssB* |
| Sequence-based reagent | TssBdel-conf-F | This paper | 5'-ATAGAAACCTAC TTTTTCGGAAAGC | Confirmation of *tssB* deletion |
| Sequence-based reagent | TssBdel-conf-R | This paper | 5'-TTACTTATTA AATAATTTATAGCTATTG AAAAGAGATAAGAATTG | Confirmation of *tssB* deletion |

*Appendix 1—key resources table continued*

| Reagent type (species) or resource | Designation | Source or reference | Identifiers | Additional information |
|---|---|---|---|---|
| Sequence-based reagent | TssMdel-1 | This paper | 5'-GATCCTCTA GAGTCGACCTGCAGGCA TG CAACCCTGTCTTGGCTA-GAGTC | Deletion of *tssM* |
| Sequence-based reagent | TssMdel-2 | This paper | 5'-ATTTGTTTTT CCGTATCAATCCAATTTCA | Deletion of *tssM* |
| Sequence-based reagent | TssMdel-3 | This paper | 5'-ATTGGATTGA TACGGAAAAACAAATATG AAAATTATTAATATTGGAG TTTTAGCTCATGTT | Deletion of *tssM* |
| Sequence-based reagent | TssMdel-4 | This paper | 5'-CTAAGTTATTTTA TTGAACATA TATCGTACTTTATCTA TCCG | Deletion of *tssM* |
| Sequence-based reagent | TssMdel-5 | This paper | 5'-AAGTACGATATATG TTCAATAAAAT AACTTAGAATAAA TTAAGGAAT TTTCAGTGCATTTGAAG | Deletion of *tssM* |
| Sequence-based reagent | TssMdel-6 | This paper | 5'-CAGGAAACA GCTATGACCATGATTACG CCGGCAATATCTAGAA CGGATTTATCG | Deletion of *tssM* |
| Sequence-based reagent | TssMdel-Conf-F | This paper | 5'-AGGACTTCC AAGATAGAAGTACGG | Confirmation of *tssM* deletion |
| Sequence-based reagent | TssMdel-Conf-R | This paper | 5'-AAAGCCCCT TGTACGATAGC | Confirmation of *tssM* deletion |
| Sequence-based reagent | Nte345del-1 | This paper | 5'-GATCCTCTA GAGTCGACCTGCAGG CATGCAGACCTTCATG CTGACTAGTGAT | Deletion of Nte3-Nte5 locus |
| Sequence-based reagent | Nte345del-2 | This paper | 5'-GAAGTGTTG GATGAACTTTTTCTATG | Deletion of Nte3-Nte5 locus |
| Sequence-based reagent | Nte345del-3 | This paper | 5'-CATAGAAAAAGTTCA TCCA ACACTTCTTAAA TTTAACGA TCACTCATCATGT | Deletion of Nte3-Nte5 locus |
| Sequence-based reagent | Nte345del-4 | This paper | 5'-TTACTTATTA AATAATTTATAGCTATTG | Deletion of Nte3-Nte5 locus |
| Sequence-based reagent | Nte345del-5 | This paper | 5'-CAATAGCTAT AAATTATTTAATAAG TAAAA TAAGAAACTGTAAACA-CAGTGTG | Deletion of Nte3-Nte5 locus |
| Sequence-based reagent | Nte345del-6 | This paper | 5'-CAGGAAACA GCTATGACCATGATTACG CCAGTTTAACTGTTC GGAAAGGGTGT | Deletion of Nte3-Nte5 locus |

*Continued on next page*

*Appendix 1—key resources table continued*

| Reagent type (species) or resource | Designation | Source or reference | Identifiers | Additional information |
|---|---|---|---|---|
| Sequence-based reagent | Nte345del-conf-F | This paper | 5'-GTTTTCGTTGG TGAGGACGG | Confirmation of Nte3-Nte5 locus deletion |
| Sequence-based reagent | Nte345del-conf-R | This paper | 5'-CTACTTATAATCCAAA TA TTTTATTGAACAGAGAAC | Confirmation of Nte3-Nte5 locus deletion |
| Sequence-based reagent | TssBsfGFP1 | This paper | 5'-CATGATTACGAATTCCC GGATTAATTAAAATGTCA CGAAACAAATCATCCGG | *tssB* amplification to fuse with *sfGFP* and clone into pNCC1-spec |
| Sequence-based reagent | TssBsfGFP2 | This paper | 5'-CTGCTCGTC TTTGGAAGC | *tssB* amplification to fuse with *sfGFP* and clone into pNCC1-spec |
| Sequence-based reagent | TssBsfGFP3 | This paper | 5'-GCTTCCAAA GACGAGCAGGCAGCAG CAGGTGGTGGTAGCAA AGGAGAAGAACTTTTCAC | *sfGFP* amplification and addition of DNA linker to fuse with *tssB* and clone into pNCC1-spec |
| Sequence-based reagent | TssBsfGFP4 | This paper | 5'-GATCCTCTA GAGTCGACCTGCAGG CATGCTCATTTGTAGA GCTCATCCATGC | *sfGFP* amplification and addition of DNA linker to fuse with *tssB* and clone into pNCC1-spec |
| Sequence-based reagent | sfGFP-Prom-F | This paper | 5'-TGACCCGGG TCATTTGTAGAGCTCA TCCATGCC | *sfGFP* amplification from pNCC1-sfGFP to clone into pNCC1-spec |
| Sequence-based reagent | sfGFP-Prom-R | This paper | 5'-TGAAAGCTTTTGACAGC TAGCTCAGTCCTAGGTATA ATGCTAGCCCAACATG TTA CACAATAATGGAGTAA TGA ACATA TGAGCAAAGGAGAAGAAC T | *sfGFP* amplification from pNCC1-*sfGFP* to clone into pNCC1-spec |
| Sequence-based reagent | pGib-RBS-Nte2-F | This paper | 5'-GATCCTCTA GAGTCGACCTGCAGGCA TG CAAAGAAGGAGATATAC- CAT GGCATTCAATAAAA TCGCCC | Nte2 amplification and addition of RBS to clone into pBAD33 |
| Sequence-based reagent | pGib-RBS-Nte2-R | This paper | 5'-AAAATCTTCTCTCA TCCGCCA AAACAGCCATCA TTTTTTCCTA TTGTTACATTTATCCT | Nte2 amplification and addition of RBS to clone into pBAD33 |
| Sequence-based reagent | pGib-RBS-Nti2-R | This paper | 5'-AAAATCTTCTCTCA TCCGCC AAAACAGCCATTATTCAAA TT TCTTTAGCAGTATTTTTCT | Nte2 and Nti2 amplification plus addition of RBS to clone into pBAD33 |

*Continued on next page*

*Appendix 1—key resources table continued*

| Reagent type (species) or resource | Designation | Source or reference | Identifiers | Additional information |
|---|---|---|---|---|
| Sequence-based reagent | pGib-RBS-Nte3-F | This paper | 5′-GATCCTCTA GAGTCGACCTGCAGG CATGCAAAGAAGGAGAT ATACCATGGCCTC TTTCGGTAAC | Nte3 amplification and addition of RBS to clone into pBAD33 |
| Sequence-based reagent | pGib-RBS-Nte3-R | This paper | 5′-AAAATCTTCTCTCA TCCGC CAAAACAGCCATCATTTAA TA CCTCTTCTTGATAATTCTTT | Nte3 amplification and addition of RBS to clone into pBAD33 |
| Sequence-based reagent | pGib-RBS-Nti3-R | This paper | 5′-AAAATCTTCTCTCA TCCGCCA AAACAGCCACTA TTCACCC AACAATGTTTCT | Nte3 and Nti3 amplification plus addition of RBS to clone into pBAD33 |
| Sequence-based reagent | PGIB-RBS-NTE4-F | This paper | 5′-GATCCTCTA GAGTCGACCTGCAGGCAT GCAAAGAAGGAGATA TACC ATGGTCGAACACAACCAG | Nte4 amplification and addition of RBS to clone into pBAD33 |
| Sequence-based reagent | pGib-RBS-Nte4-R | This paper | 5′-AAAATCTTCTCTCA TCCGC CAAAACAGCCATTAAATTA TT GGAAGATTTTTACAACCA | Nte4 amplification and addition of RBS to clone into pBAD33 |
| Sequence-based reagent | pGib-RBS-Nti4-R | This paper | 5′-AAAATCTTCTCTCA TCCGCC AAAACAGCCATTACGCTT TTAAATTCCGGTG | Nte4 and Nti4 amplification plus addition of RBS to clone into pBAD33 |
| Sequence-based reagent | pGib-RBS-Nte5-F | This paper | 5′-GATCCTCTA GAGTCGACCTGCAGGCAT GCAAAGAAGGAGATATAC CATGGGTCGTC TGAAAAGC | Nte5 amplification and addition of RBS to clone into pBAD33 |
| Sequence-based reagent | pGib-RBS-Nte5-R | This paper | 5′-AAAATCTTCTCTCA TCCGC CAAAACAGCCACTAATC TA ATCGTTTGGGCG | Nte5 amplification and addition of RBS to clone into pBAD33 |
| Sequence-based reagent | pGib-RBS-Nti5-R | This paper | 5′-AAAATCTTCTCTCA TCCG CCAAAACAGCCATTAA TCC CAATAACTGTCTAAATTGT | Nte5 and Nti5 amplification plus addition of RBS to clone into pBAD33 |
| Sequence-based reagent | pGib-RBS-Nte6-F | This paper | 5′-GATCCTCTA GAGTCGACCTGCAGGC ATGCAAAGAAGGAGATA TACCATGGCCTCTTTCGG TAAC | Nte6 amplification and addition of RBS to clone into pBAD33 |
| Sequence-based reagent | pGib-RBS-Nte6-R | This paper | 5′-AAAATCTTCTCTCATCC GCCAAAACAGCCACTA TTA TCTAGGAACAATCTGAT TAATTATTCC | Nte6 amplification and addition of RBS to clone into pBAD33 |

*Continued on next page*

*Appendix 1—key resources table continued*

| Reagent type (species) or resource | Designation | Source or reference | Identifiers | Additional information |
|---|---|---|---|---|
| Sequence-based reagent | pGib-RBS-Nti6-R | This paper | 5'-AAAATCTTCTCTCATCCGCCAAAACAGCCATTAAAT TTCCTCTAGTTTTTCTTTCATC | Nte6 and Nti6 amplification plus addition of RBS to clone into pBAD33 |
| Sequence-based reagent | CE043-F | This paper | 5'-GGCCGGTCTAGAAAGAAGGAGATATACCATGAAATACCTGCTGCCGACCGCTGCTGCTGGTCTGCTGCTCCTCGC | Addition of 5' PelB leader peptide and 3' 6xHIS-tag to PLA2 domain to clone into pBAD33 |
| Sequence-based reagent | CE044-F | This paper | 5'-GGTCTGCTGCTCCTCGCTGCCCAGCCGGCGATGGCCATGGGGGGAAGTAATTTTTATGCGTTTGCA | PLA2 domain amplification and addition of 5' PelB leader peptide |
| Sequence-based reagent | CE046-R | This paper | 5'-CCGGCCGCATGCCTAGTGATGGTGATGGTGATGCCTATGATTTTTA-GAC | Addition of 3' 6xHIS-tag to PLA2 domain with or without 5' PelB leader peptide to clone into pBAD33 |
| Sequence-based reagent | CE047-R | This paper | 5'-GATGCCTATGATTTTTAGACGTTTTTTTAATTGTTTTATCG | PLA2 domain amplification with or without addition of 5' PelB leader peptide |
| Sequence-based reagent | CE048-R | This paper | 5'-CCGGCCGCATGCCTAGTGATGGTGATGGTGATGATTAAGTTTGGATAGTTTGAAAATTTTTTTAAGCTTATATATAAG | PLA2 domain amplification with or without a 5' PelB leader peptide and amplification of Nti1 adding a 3' 6xHIS-tag to clone into pBAD33 |
| Sequence-based reagent | CE083-F | This paper | 5'-GGCCGGTCTAGAAAGAAGGAGATATACCATGGGGGGGAAGTAATTTTTATGCGTTTGCA | PLA2 domain amplification and addition of 3' 6xHIS-tag to clone into pBAD33 |
| Sequence-based reagent | MW312 | This paper | 5'- TATAAGGAGGAACATATGGAATACATGTTATAATAACTATAAC | Spectinomycin cassette amplification from pDG1728 to clone into pNCC1 |
| Sequence-based reagent | MW313 | This paper | 5'- GTATTCCATATGTTCCTCCTTATAAAATTAGTATAATTATAG | pNCC1 plasmid backbone amplification |
| Sequence-based reagent | MW314 | This paper | 5'- GCATCCCTTAACGACGTCAATTGAAAAAAGTGTTTCCACC | Spectinomycin cassette amplification from pDG1728 to clone into pNCC1 |
| Sequence-based reagent | MW315 | This paper | 5'-TCAATTGACGTCGTTAAGGGATGCATAAACTGCATCCCTTAAC | pNCC1 plasmid backbone amplification |

