## [Decision Letter]

**Acceptance summary:**

The type VI secretion system (T6SS) is an antibacterial system that has been characterized in a large number of Gram-negative bacteria either pathogenic or environmental. This study represents the first functional description of T6SS in the genus *Neisseria*, which, unusually, is plasmid encoded. The work demonstrates that this T6SS may be important in allowing commensal *Neisseria cinerea* to resist pathogenic *Neisseria,* and has implications for understanding how spatial considerations impact T6SS-mediated inter-bacterial interactions.

**Decision letter after peer review:**

Thank you for submitting your article "Type VI secretion system killing by commensal Neisseria is influenced by the spatial dynamics of bacteria" for consideration by *eLife*. Your article has been reviewed by 3 peer reviewers, including Alain Filloux as the Reviewing Editor and Reviewer #1, and the evaluation has been overseen by Gisela Storz as the Senior Editor.

The reviewers have discussed the reviews with one another and the Reviewing Editor has drafted this decision to help you prepare a revised submission.

We would like to draw your attention to changes in our policy on revisions we have made in response to COVID-19 (https://elifesciences.org/articles/57162). Specifically, when editors judge that a submitted work as a whole belongs in *eLife* but that some conclusions require a modest amount of additional new data, as they do with your paper, we are asking that the manuscript be revised to either limit claims to those supported by data in hand, or to explicitly state that the relevant conclusions require additional supporting data.

Summary:

The type VI secretion system (T6SS) is an antibacterial system that has been characterized in a large number of Gram-negative bacteria either pathogenic or environmental.

This works represents the first functional description of T6SS in the genus Neisseria, which, unusually, is plasmid encoded, demonstrates that this T6SS may be important in allowing commensal Neisseria cinerea to resist pathogenic Neisseria, assess the role of T6SS effectors and how the capsule on the target cell reduces the efficacy of T6SS killing.

It brings important new aspects to the emerging theme in the field that spatial considerations are critical to understanding T6SS-mediated inter-bacterial interactions. Notably, it demonstrates how the type four pilus (Tfp) on the target cell contributes to T6SS-dependent killing. In a mixed population of bacteria possessing or not the Tfp, bacteria without Tfp are excluded at the edge of the microcolony where they can outgrow suggesting that both T6SS and Tfp contribute to the spatial organisation of the microcolonies.

Overall, the findings will be of great interest to the T6SS field and also those interested in microbial interactions more broadly.

Essential revisions:

1. The focus of the paper is on the spatial dynamics/spatial segregration in mixed microcolonies. Yet, the authors do not exclude a very localized role of the Tfp at the interface between the attacker and the prey. It is not clear for me whether the impact of the Tfp is due to organisation of the bacterial community (with Tfp minus strains segregating at the periphery of the colony) or whether the Tfp has a very localized impact either by bringing the attacker and the prey in close contact or triggering a response of the attacker as described for vibrio T4SS/mating pair formation system. Indeed, in figure 6, it seems that there are also major differences in the center of the inoculation zone between Tfp- and Tfp+ strains.

Assessing the role of Tfp in the attacker cells (in the experiment presented in Figure 6) would be very helpful for the understanding of the role of Tfp.

2. There are 6 toxin/immunity pairs described. While Nte1-5 are encoded in the T6SS cluster, nte6/nti6 is found as a remote pair. What suggests that Nte6 is a T6SS-dependent toxin? In this respect there are no demonstration of a direct T6SS-dependent secretion of any of the 6 toxins. This is not essential but if tools available and at least one toxin could be tested it will add value to this publication.

3. There does not seem to be any paar gene in the cluster. Is there any elsewhere on the plasmid? If not which of the Ntes have a PAAR-like at the N terminus. Only Nte1 and Nte2? If this is true, then deletion of these two should likely prevent the other toxins to be delivered since it is accepted that at least one PAAR is required for the T6SS to be effective. This may be worth assessing but at least some more information on the domain organisation would be helpful. Along this line, the Nte5 (and to a lesser extent Nte4) effector is much smaller than a typical full-length T6SS-associated Rhs protein would be. Is it (or both) actually an 'orphan' Rhs CT- immunity pair (presumed to be displaced from the upstream full-length Rhs by homologous recombination, leaving only a part of the Rhs domain)? Perhaps this could be discussed?

[Editors' note: further revisions were suggested prior to acceptance, as described below.]

Thank you for resubmitting your work entitled "Type VI secretion system killing by commensal Neisseria is influenced by the spatial dynamics of bacteria" for further consideration by *eLife*. Your revised article has been evaluated by Gisela Storz as the Senior Editor, and a Reviewing Editor.

All previous reviewers have had the opportunity to look carefully at the changes you made and we are happy to say that there is a consensus that most comments have been well addressed. However, there is still the pending issue that the contribution of type IV pili (Tfp) loss to increased prey survival may not simply be due to the gross spatial reorganisation in the colony. One way to look a bit further into this would be for example to have attacker and prey both lacking pili so that the role of Tfp in T6SS-dependent combat independently of segregation could be further tested. One reviewer suggests for example an experiment where you make one more strain (GFP Tfp- strain to be used as T6SS+/Tfp- attacker) and then use this with the Tfp+ and Tfp- prey cells (Tfp- attacker with Tfp- prey would give a non-segregated population but might retain other aspects of loss of Tfp, so would be interesting to see if prey still survive better/T6SS still less efficient).

The *eLife* policy is not to ask for a second round of revision. However, in this case we thought we could give you the opportunity to address this point, especially since we felt that it should not be a too challenging experiment.

Will you decide not to perform the experiment then we would ask you to modify your text accordingly so that the discussion about the role of Tfp takes all possible aspects into consideration. We would also suggest to modify the title to highlight the role of the Tfp without any emphasis on the spatial dynamics."

---

## [Author Response]

Essential revisions:1. The focus of the paper is on the spatial dynamics/spatial segregration in mixed microcolonies. Yet, the authors do not exclude a very localized role of the Tfp at the interface between the attacker and the prey. It is not clear for me whether the impact of the Tfp is due to organisation of the bacterial community (with Tfp minus strains segregating at the periphery of the colony) or whether the Tfp has a very localized impact either by bringing the attacker and the prey in close contact or triggering a response of the attacker as described for vibrio T4SS/mating pair formation system. Indeed, in figure 6, it seems that there are also major differences in the center of the inoculation zone between Tfp- and Tfp+ strains.Assessing the role of Tfp in the attacker cells (in the experiment presented in Figure 6) would be very helpful for the understanding of the role of Tfp.

We agree with the reviewer that Tfp may have a localised impact through pilus-pilus interactions bringing individual cells into close proximity for T6SS dependent killing. However, the assays presented in this paper do not enable us to either ascertain or exclude this effect. Further experiments (*e.g*. single cell analysis of piliated and non-piliated attacker/prey using sytox blue as an indicator of prey lysis) are required to address this but we feel are beyond the scope of this work. We have modified our discussion to clarify that our current work is limited to understanding the impact of prey piliation of a susceptible prey on survival against T6SS attack within a bacterial community, and that the contribution of pili on the attacker and at the cell-cell level merits future investigation

In addition, analysis of the impact of pili on attacker and the impact of pilin subunit variation, Tfp dynamics and the contribution of effectors should be addressed in future work to provide a comprehensive view of the interplay between Tfp expression and T6SS attack in mixed microcolonies.

2. There are 6 toxin/immunity pairs described. While Nte1-5 are encoded in the T6SS cluster, nte6/nti6 is found as a remote pair. What suggests that Nte6 is a T6SS-dependent toxin? In this respect there are no demonstration of a direct T6SS-dependent secretion of any of the 6 toxins. This is not essential but if tools available and at least one toxin could be tested it will add value to this publication.

The reviewer raises a valid point. Our reasons for proposing Nte6 as a T6SS-dependent toxin were based on bioinformatic analysis which reveals that the protein encoded by Nte6 has features commonly found in T6SS effectors (namely Rhs domains and a C-terminal region encoding a toxin belonging to the HNH superfamily, see revised manuscript Figure 4). In addition, some preliminary data we have obtained indicates this effector is linked with the T6SS. As shown in Author response image 1, deletion of *nte6* in combination with the three other genes encoding putative nuclease effectors (generating 346T*Δnte/nti3,4,5,6*) led to survival of prey (*N. meningitidis*) at levels equivalent to survival in presence of a ΔT6SS mutant, and abolished Hcp secretion. This observation suggests that Nte3, Nte4, Nte5 and Nte6 are linked to the T6SS in *N. cinerea* and may be required for T6SS activity, reminiscent of findings from *Agrobacterium tumefaciens* in which effective T6SS activity requires loading of cargo effectors (Wu et al., 2020). However, these findings remain preliminary and experimental data demonstrating T6SS-dependent secretion of Nte1-6 in *N. cinerea* 346T are still required. Therefore, we have been careful to describe these as putative effectors in our current manuscript, and to clarify that further experiments are required to demonstrate direct T6SS-dependent secretion for all the toxins .

**Author response image 1. sa2fig1:** The deletion of all four putative endonuclease effectors impairs T6SS function and Hcp secretion. (A) Recovery of wild-type *N. meningitidis* 8013 after 4 h co-incubation with *N. cinerea* 346T wild-type (Wt) and the T6SS mutant (ΔT6SS) or strains lacking specific effectors at a 100:1 attacker:prey ratio. Data shown are the mean ± SD of three independent experiments: NS, not significant, ***p < 0.0001, one-way ANOVA test for multiple comparison. (B) Western blot detection of Hcp in whole cell lysates (W) and supernatants (S) from bacteria grown in liquid BHI media for 4-5 h. WT (wild-type *N. cinerea* 346T), ΔTssB (strain with deletion of *tssB*), Δ4Nte (strain lacking *nte/nti 3,4,5,6*).

3. There does not seem to be any paar gene in the cluster. Is there any elsewhere on the plasmid? If not which of the Ntes have a PAAR-like at the N terminus. Only Nte1 and Nte2? If this is true, then deletion of these two should likely prevent the other toxins to be delivered since it is accepted that at least one PAAR is required for the T6SS to be effective. This may be worth assessing but at least some more information on the domain organisation would be helpful.

The reviewer is correct, only Nte1 and Nte2 have PAAR domains at the N terminus (indicated in new Figure 4 ). We analysed the PacBio WGS of *N. cinerea* 346T for additional PAAR domains using BLASTp analysis and the PAAR-like domain sequence from Nte1 as query sequence. This analysis did not identify any other PAAR genes either on the plasmid or the chromosome. We have now included this information in the revised manuscript .

The reviewer is correct that based on previous work, deletion of both PAAR containing effectors (Nte1 and Nte2) may prevent secretion, and this is something we intend to address in future work, along with demonstration of the T6SS-dependent secretion of the toxins as indicated in response to point 2. As suggested, our revised manuscript includes more information about the predicted effectors (new Figure 4- schematic of domain structure, results , discussion and Materials and methods ).

Along this line, the Nte5 (and to a lesser extent Nte4) effector is much smaller than a typical full-length T6SS-associated Rhs protein would be. Is it (or both) actually an 'orphan' Rhs CT- immunity pair (presumed to be displaced from the upstream full-length Rhs by homologous recombination, leaving only a part of the Rhs domain)? Perhaps this could be discussed?

The reviewer raises and interesting point. Orphan effector-immunity gene pairs have been reported, for example in *Vibrio* species, where they are found in conserved gene neighbourhoods, not adjacent to the T6SS gene cluster and sometimes accompanied by transposable elements (Salomon et al., 2015). In addition, orphan immunity genes have been identified in several species (Ma et al., 2017; Ross et al., 2019) and can be found as islets, or located downstream of effector-immunity modules (Kirchberger et al., 2017). Nte5 encodes a putative nuclease effector with a shorter Rhs domain compared to the other five putative effectors (319AA compared to 695-735AA) and therefore may represent an orphan RHS-CT gene. We have now included this possibility in the revised manuscript .

References:

Custodio, R., Johnson, E., Liu, G., Tang, C.M., and Exley, R.M. (2020). Commensal *Neisseria cinerea* impairs *Neisseria meningitidis* microcolony development and reduces pathogen colonisation of epithelial cells. PLoS pathogens 16, e1008372.

Hagblom, P., Segal, E., Billyard, E., and So, M. (1985). Intragenic recombination leads to pilus antigenic variation in *Neisseria gonorrhoeae*. Nature 315, 156-158.

Helm, R.A., and Seifert, H.S. (2010). Frequency and rate of pilin antigenic variation of *Neisseria meningitidis*. J Bacteriol 192, 3822-3823.

Kirchberger, P.C., Unterweger, D., Provenzano, D., Pukatzki, S., and Boucher, Y. (2017). Sequential displacement of Type VI Secretion System effector genes leads to evolution of diverse immunity gene arrays in *Vibrio cholerae*. Scientific reports 7, 45133.

Knapp, J.S., and Hook, E.W., 3rd (1988). Prevalence and persistence of *Neisseria cinerea* and other *Neisseria* spp. in adults. J Clin Microbiol 26, 896-900.

Ma, J., Pan, Z., Huang, J., Sun, M., Lu, C., and Yao, H. (2017). The Hcp proteins fused with diverse extended-toxin domains represent a novel pattern of antibacterial effectors in type VI secretion systems. Virulence 8, 1189-1202.

Marri, P.R., Paniscus, M., Weyand, N.J., Rendon, M.A., Calton, C.M., Hernandez, D.R., Higashi, D.L., Sodergren, E., Weinstock, G.M., Rounsley, S.D., et al. (2010). Genome sequencing reveals widespread virulence gene exchange among human *Neisseria* species. PLoS One 5, e11835.

Matthey, N., Stutzmann, S., Stoudmann, C., Guex, N., Iseli, C., and Blokesch, M. (2019). Neighbor predation linked to natural competence fosters the transfer of large genomic regions in *Vibrio cholerae*. *eLife* 8.

Ross, B.D., Verster, A.J., Radey, M.C., Schmidtke, D.T., Pope, C.E., Hoffman, L.R., Hajjar, A.M., Peterson, S.B., Borenstein, E., and Mougous, J.D. (2019). Human gut bacteria contain acquired interbacterial defence systems. Nature 575, 224-228.

Salomon, D., Klimko, J.A., Trudgian, D.C., Kinch, L.N., Grishin, N.V., Mirzaei, H., and Orth, K. (2015). Type VI Secretion System Toxins Horizontally Shared between Marine Bacteria. PLoS pathogens 11, e1005128.

Sheikhi, R., Amin, M., Rostami, S., Shoja, S., and Ebrahimi, N. (2015). Oropharyngeal Colonization With *Neisseria lactamica*, Other Nonpathogenic *Neisseria* Species and *Moraxella catarrhalis* Among Young Healthy Children in Ahvaz, Iran. Jundishapur J Microb *8*.

Wu, C.F., Lien, Y.W., Bondage, D., Lin, J.S., Pilhofer, M., Shih, Y.L., Chang, J.H., and Lai, E.M. (2020). Effector loading onto the VgrG carrier activates type VI secretion system assembly. EMBO reports 21, e47961.

[Editors' note: further revisions were suggested prior to acceptance, as described below.]

All previous reviewers have had the opportunity to look carefully at the changes you made and we are happy to say that there is a consensus that most comments have been well addressed. However, there is still the pending issue that the contribution of type IV pili (Tfp) loss to increased prey survival may not simply be due to the gross spatial reorganisation in the colony. One way to look a bit further into this would be for example to have attacker and prey both lacking pili so that the role of Tfp in T6SS-dependent combat independently of segregation could be further tested. One reviewer suggests for example an experiment where you make one more strain (GFP Tfp- strain to be used as T6SS+/Tfp- attacker) and then use this with the Tfp+ and Tfp- prey cells (Tfp- attacker with Tfp- prey would give a non-segregated population but might retain other aspects of loss of Tfp, so would be interesting to see if prey still survive better/T6SS still less efficient).The eLife policy is not to ask for a second round of revision. However, in this case we thought we could give you the opportunity to address this point, especially since we felt that it should not be a too challenging experiment.

We are very grateful to the reviewer for this suggestion. We had available the non-piliated, T6SS+ strain expressing GFP (included in revised Key resources table) and have now performed this experiment using (i) competition assays to quantitatively assess prey survival and (ii) fluorescence microscopy to qualitatively assess the distribution of strains in mixed colonies. The competition assays revealed that increased prey survival is observed only when the prey is non-piliated, thus, these data support our conclusions that prey survival is enhanced upon loss of pili due to segregation. However, although competition assays demonstrated that prey survival was equivalent when attacker and prey either both have, or both lack Tfp, the microscopy analysis of mixed colonies indicate qualitative differences in prey distribution between colonies composed of Tfp+/Tfp+ attacker and prey and Tfp-/Tfp- attacker and prey. These observations suggests that Tfp may have an effect beyond gross spatial reorganisation, possibly a local effect that is not detected at the population level. We have included the new data and discussion of these findings in our revised manuscript (Figure 8, Figure 8—figure supplement 1 and 2 and corresponding legends, results , discussion).

Will you decide not to perform the experiment then we would ask you to modify your text accordingly so that the discussion about the role of Tfp takes all possible aspects into consideration. We would also suggest to modify the title to highlight the role of the Tfp without any emphasis on the spatial dynamics."

We have performed the experiment and have also modified the text accordingly. As our findings do not rule out contributions of Tfp other than Tfp loss enhancing prey survival through segregation, we have included other possible roles of Tfp in the results and discussion and still include that further work is necessary to explore the contribution of Tfp to T6SS mediated attack beyond the impact on spatial reorganisation within a colony .

Regarding the title of our paper, we felt that the reviewers’ suggestion was helpful even though we have included further data to support the observation that T6SS killing in *N. cinerea* is influenced by spatial dynamics. We have therefore modified the title to be a broader description of the work presented. The new title is “Type VI secretion system killing by commensal *Neisseria* is influenced by expression of Type four pili” and highlights the role of Tfp without any emphasis on the spatial dynamics.